# DNA methylation as a pharmacodynamic marker of glucocorticoid response and glioma survival

J. K. Wiencke [1] ✉, Annette M. Molinaro [1], Gayathri Warrier [1], Terri Rice[1], Jennifer Clarke [1,2], Jennie W. Taylor [1,2], Margaret Wrensch[1], Helen Hansen [1], Lucie McCoy [1], Emily Tang [3], Stan J. Tamaki[4], Courtney M. Tamaki[4], Emily Nissen [5], Paige Bracci[3], Lucas A. Salas [6], Devin C. Koestler[5], Brock C. Christensen[6,7,8], Ze Zhang [6] & Karl T. Kelsey [9,10]

Assessing individual responses to glucocorticoid drug therapies that compromise immune status and affect survival outcomes in neuro-oncology is a great challenge. Here we introduce a blood-based neutrophil dexamethasone methylation index (NDMI) that provides a measure of the epigenetic response of subjects to dexamethasone. This marker outperforms conventional approaches based on leukocyte composition as a marker of glucocorticoid response. The NDMI is associated with low CD4 T cells and the accumulation of monocytic myeloid-derived suppressor cells and also serves as prognostic factor in glioma survival. In a non-glioma population, the NDMI increases with a history of prednisone use. Therefore, it may also be informative in other conditions where glucocorticoids are employed. We conclude that DNA methylation remodeling within the peripheral immune compartment is a rich source of clinically relevant markers of glucocorticoid response.

Cortisol and synthetic glucocorticoid analogs activate intracellular glucocorticoid receptors (GR) and modulate many physiologic processes. Processes affected include carbohydrate and lipid metabolism, stress, and immune responses[1]. Because of their potent effects on immune cells, glucocorticoids have long been the most widely prescribed class of anti-inflammatory and immunosuppressive drugs[2]. However, glucocorticoid-treated patients may exhibit resistance or experience a wide range of adverse side effects. Our understanding of these risks is incomplete[3,4]. To account for individual variability, biomarkers have been developed that integrate variations in glucocorticoid exposure and response[5]. Several well-established safety concerns

of glucocorticoids have been addressed with markers bridged to clinical outcomes, namely: adrenal suppression, insulin resistance, and accelerated bone turnover[6–9]. In the management of brain tumor patients, the synthetic glucocorticoid dexamethasone (DEX) is a mainstay for control of peritumoral edema, helping to alleviate symptoms due to high intracranial pressure or presence of tumor/edema in eloquent areas of brain[10]. No randomized controlled trials have specifically addressed DEX. However, its' use in glioma patients has been associated with poor survival[11–13] and compromised response to immunotherapy[14–16], There are no established markers to assess these risks, although altered blood leukocyte composition has been

[1]Department of Neurological Surgery, University of California San Francisco, San Francisco, CA, USA. [2]Department of Neurology, University of California San Francisco, San Francisco, CA, USA. [3]Department of Epidemiology and Biostatistics, University of California San Francisco, San Francisco, CA, USA. [4]Parnassus Flow Cytometry CoLab, University of California San Francisco, San Francisco, CA, USA. [5]Department of Biostatistics & Data Science, University of Kansas Medical Center, Kansas City, KS, USA. [6]Department of Epidemiology, Geisel School of Medicine, Dartmouth College, Lebanon, NH, USA. [7]Department of Molecular and Systems Biology, Geisel School of Medicine, Dartmouth College, Lebanon, NH, USA. [8]Department of Community and Family Medicine, Geisel School of Medicine, Dartmouth College, Lebanon, NH, USA. [9]Department of Epidemiology, Brown University, Providence, RI, USA. [10]Department of Pathology and Laboratory Medicine, Brown University, Providence, RI, USA. ✉e-mail: John.Wiencke@ucsf.edu

used to mark glucocorticoid-related immunosuppression[17–19]. Depletion of circulating lymphocyte numbers[20–23] and expansion of neutrophils and immunosuppressive myeloid cell populations[24,25], including monocytic myeloid-derived suppressor cells (mMDSC)[26,27], have been documented in DEX exposed brain tumor patients. Although blood leukocyte compositions are sensitive to glucocorticoid exposures, they lack specificity, as tumor inflammation and chemoradiation also directly affect them[28]. In our search to uncover more specific markers of glucocorticoid response, we turned to recent epigenetic studies showing that cell-specific chromatin modifications play a pivotal role in determining the genomic effects of glucocorticoids[29,30]. Cell-specific actions of glucocorticoids have been linked to highly accessible chromatin domains located preferentially within distal enhancer elements that are intrinsic to cell lineage[31]. Glucocorticoid alterations of DNA methylation may be integral to their genomic effects[32,33]. Because DNA methylation is a chemically stable endpoint and easy to access in peripheral blood, it is an attractive potential marker for glucocorticoid pathway activation. Previous studies of DNA methylation and glucocorticoids focused on a limited number of genes and cytosine guanine dinucleotide (CpG) loci in the context of neurodevelopmental and stress-related outcomes[34,35]. Furthermore, prior research did not take advantage of the cell-selective nature of glucocorticoid-mediated genomic effects. Comparative gene expression studies showed that, among different hematopoietic cell types, neutrophils exhibited the most pronounced transcriptional response to glucocorticoid treatments[36]. In severe COVID-19 subjects DEX induced profound transcriptional remodeling and induction of immunosuppressive neutrophil states[37].

In this work, we explore DNA methylation as pharmacodynamic markers of DEX exposure in adult glioma patients who had and had not been given the drug as part of their routine care. We employ a bioinformatic approach to discover and assess DEX-related neutrophil-specific DNA methylation as a marker of in vivo glucocorticoid exposure and response, the neutrophil dexamethasone methylation index (NDMI). Our results indicate that blood DNA methylation can accurately discriminate DEX exposed and non-exposed subjects and that a methylation marker targeting blood neutrophils performs better than a marker based on altered blood leukocyte composition in characterizing individual glucocorticoid response. We also explore the association of NDMI scores with blood counts of CD4 T cells and mMDSC and show that neutrophil-specific methylation markers of DEX exposure serve as prognostic factors in brain tumor survival.

## Results

### Neutrophil-specific and non-cell-specific DNA methylation induced by DEX

Clinical and demographic characteristics of our study groups and study design are presented (Fig. 1a; Supplementary Table 1). After modeling all cell types using CellDMC, we identified 2621 CpG probes that displayed statistically significant ($p < 0.01$) interactions with neutrophil proportions and one CpG that indicated significant interaction ($p < 0.01$) with B-cell proportions (Figs. 1b, 2a, Supplementary Data File 1). The predominant effect of DEX was hypomethylation. Another subset of 17,733 loci was affected by DEX but not in a cell-specific fashion (i.e., non-cell-specific methylation).

### Genomic and functional features of neutrophilic DEX responsive loci

To test whether neutrophil-specific CpGs follow pre-established and lineage-specific DNA methylation patterns, we explored the methylation levels of DEX-responsive loci in isolated non-exposed neutrophils and other immune cell types. DEX-related neutrophil loci displayed intermediate levels of methylation. These loci were partially demethylated in unexposed neutrophils and other myeloid cells (basophils, eosinophils, monocytes), whereas, in lymphocytic cell types (CD4 T,

CD8 T, NK cell), they were hypermethylated (Fig. 2b). Interestingly, some B-cell loci were partially demethylated, which appeared related to their naïve and memory differentiation status. The average methylation beta values of DEX loci in neutrophils were markedly lower than lymphoid cells. Similarly, other myeloid cells (eosinophils, basophils, and monocytes) demonstrated lower beta values than lymphocytes (Fig. 2c).

Mapping the genomic locations of neutrophil-specific loci revealed overlap at GR binding sites and occurred preferentially within introns, enhancers, and in a non-CpG island context (Fig. 3a). Gene ontology (GO) analyses indicate enrichment among 100 genes for neutrophil activation and granule biology (Fig. 3b). Comparing these enriched GO-associated genes with published 11 RNA co-expression modules (EM1-11) in human bone marrow myelopoiesis[38] revealed a close correspondence of 21 differentially methylated genes with one co-expression module (designated EM6 Fig. 3c) that was maximally transcribed at the metamyelocyte and band stages (Fig. 3d).

### Machine learning approach to DEX predictors using leukocyte composition and DNA methylation

DEX associated alterations in blood leukocyte composition in the training dataset followed expectations based on historical experience with glucocorticoids: increases in neutrophil proportions and decreases in lymphocytes were observed in DEX exposed subjects (Supplementary Table 3). Although absolute monocyte counts were marginally increased, monocyte proportions were not significantly higher in DEX exposed subjects. Elastic net regularized models were fit using all six cell proportions and the ratios of neutrophil/lymphocytes, CD4/CD8 T cells, and lymphocytes/monocytes for DEX exposed and non-exposed subjects. The resulting blood leukocyte composition index contains five parameters (neutrophil, monocyte, NK and CD4 T-cell proportions, and neutrophil/lymphocyte ratio), along with their regression coefficients. Elastic net models were also constructed using the 2621 neutrophil-specific and 17,733 non-cell-specific methylation loci. The elastic net model fit to the 2621 neutrophil-specific DEX associated loci resulted in 28 CpGs with non-zero coefficients. A classifier, termed the neutrophil dexamethasone methylation index (NDMI), was created based on these 28 CpGs and is defined as follows:

$$p = Probability\ of\ being\ a\ Dex\ User = \frac{1}{1 + e^{-NDMI}} \quad (1)$$

$$NDMI = \beta_0 + (\beta_1 * (cg00052684) + (\beta_2 * (cg01994208) + \cdots\cdots + (\beta_{28} * (cg27094376)) \quad (2)$$

The NDMI contains well-known glucocorticoid targets of demethylation such as *FKBP5* and *ZBTB16* and others with less well-described roles in glucocorticoid response (Supplementary Table 4). Entering the 28 sites into the eFORGE program revealed a significant over-representation of the enhancer chromatin mark H3K1me3 in primary human monocytes but not in T or B cells or other hematopoietic cells; neutrophils are not represented in the eFORGE reference database (Supplementary Fig. 1). NDMI loci were often sites (16/28) for transcription factor binding, including Ikaros (IKZF1, IKZF2), MYC, and MAX, which can form multi-protein repressive complexes (Supplementary Table 7, Supplementary Data File 2). In contrast to the larger set of neutrophil genes identified in GO analysis, the 16 genes represented in the NDMI do not overlap with the EM6 bone marrow co-expression module and instead display maximal transcription during the earliest and late stages of neutrophil maturation (Fig. 3e). The predictor based on 61 non-cell-specific CpGs is termed the non-cell-specific predictor. The components of each predictor are summarized (Supplementary Table 5).

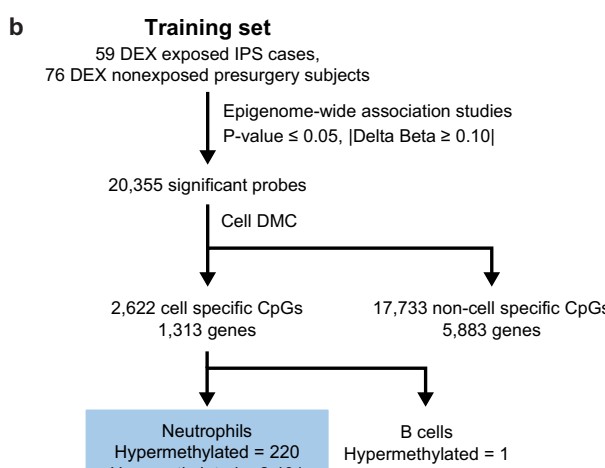

**Sources of variation in NDMI scores include age, a history of prednisone use and baseline NDMI status**

We estimated potential technical error in NDMI measurements using replicate methylation measures from 17 subjects (coefficient of variation <18% see Supplementary Table 7). We found age to be positively associated with NDMI scores of the demographic factors. The current use of glucocorticoid medications was associated with higher NDMI scores (Supplementary Table 8, Supplementary Fig. 2). Oral prednisone users had higher NDMI scores than non-users ($P = 0.003$) or subjects using inhaled glucocorticoids ($P = 0.03$). NDMI scores of inhaled glucocorticoid users were not statistically significantly different from non-glucocorticoid users. Medical conditions recorded among the 18 non-glioma controls receiving oral prednisone included rheumatoid arthritis, solid organ transplants, kidney disease, lupus,

**Fig. 1 | Study design and results. a** There were four phases to the design. First, in Biomarker Discovery two pipelines (EWAS/CellDMC (see **b**) and immune cell deconvolution) were applied to a cohort of 135 immune profile study (IPS) glioma cases of which 59 were DEX exposed and 76 were not. Second, three different elastic net regression models (DEX Exposure Predictors) were built using different CpGs (Non-cell-specific (1st model) and Neutrophil specific (NDMI: 2nd model)) and 5 immune parameters (3rd model) on the same 135 IPS glioma cases. Third, the three models were evaluated by an independent cohort of 552 Adult Glioma Study (AGS) cases and controls. Fourth, the NDMI model was evaluated as a predictor for survival in 429 AGS glioma cases of which 172 were exposed to DEX and 257 were not. **b** Overview of CpGs selected from EWAS/CellDMC pipeline in Biomarker Discovery (1st phase of **a**). Nominally significant loci ($p < 0.05$) were identified and entered into the CellDMC program resulting in loci with significant interaction terms (FDR < 0.05). Two subsets of loci were identified non-cell specific and Neutrophil specific (1st and 2nd models in **a**). Source data are provided as a Source Data file.

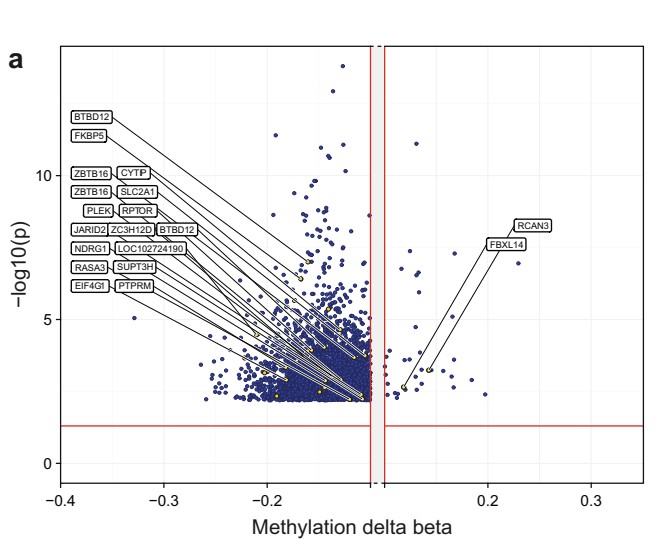

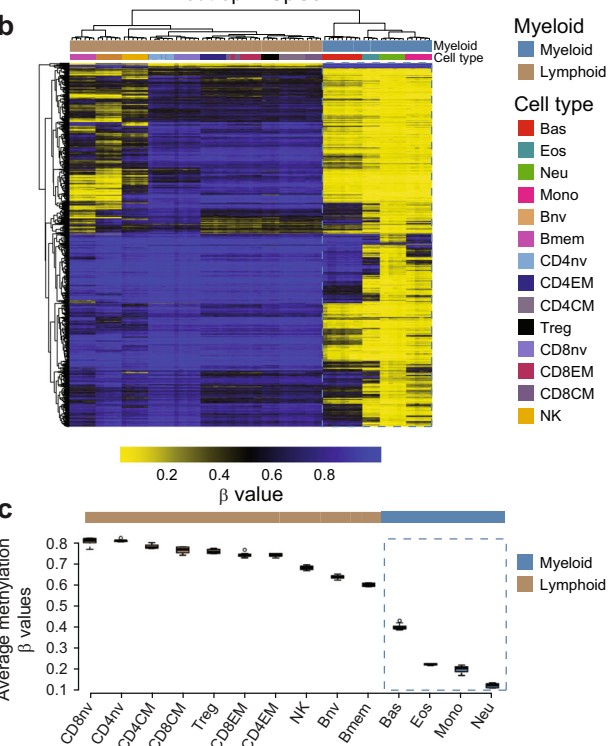

**Fig. 2 | DEX-related neutrophil-specific DNA methylation. a** Volcano plot showing the output of CellDMC analysis of 2621 DEX associated CpG loci with nominally significant (FDR < 0.05) interactions with neutrophil proportions. Plotted are the average difference in methylation beta value (delta beta) between DEX exposed and DEX non-exposed subjects (x-axis) and the $p$ value for the interaction of methylation and neutrophil proportion (y-axis). Hypomethylated loci have lower methylation beta values in exposed versus non-exposed (left side of volcano plot). The plot is truncated at |delta beta| > 0.1. CpGs associated with genes included in the NDMI score are highlighted. **b** Heat map of methylation beta values of 2621 DEX associated neutrophil probes (output of CellDMC) across 14 different isolated non-exposed immune cell types. Median beta values and Tukey's interference (inter-quartile range; Q3–Q1) are depicted; 6 individuals were sampled for CD4nv, CD8nv, CD4CM, CD8CM, CD4EM, CD8EM, NK, Bnv, Bmem, mono, neutrophils, and 9 basophil, 4 Treg, 4 Eos, 3 monocytes. Myeloid cells are demarcated with a dashed line. **c** The median beta values and Tukey's interference range (interquartile range) are depicted for each of 14 immune cell types contained in **b**; note cells are ordered from highest to lowest beta value. Source data are provided as a Source Data file.

scleroderma, multiple sclerosis, uveitis, hepatitis, and connective tissue disorder. The changes in immune cell proportion in controls taking prednisone were similar to those observed among glioma patients treated with DEX, Supplementary Table 9. Among the DEX exposed glioma subjects in the training data, cumulative and average daily mg DEX exposure was associated with variations in the NDMI; dose–response plots suggested a non-linear and saturating relationship between NDMI score and DEX dose (Fig. 4a, b). Several DEX users had lower scores overlapping non-exposed cases. These outliers included 7 DEX users with low cumulative doses and recent (within 30 days) but not current DEX exposure, suggesting a time-dependent change in NDMI following cessation of DEX treatment. In a longitudinal analysis of subjects who completed DEX treatments before the blood draw, we observed four subjects with no exposure for seven or more days before the blood draw had NDMI scores identical to their pre-treatment scores (Supplementary Fig. 3). In another longitudinal cohort of 31 DEX naïve subjects subsequently treated with DEX following surgery, the NDMI scores increased during DEX exposure (i.e.,

at blood draw) and were inversely correlated with each subject's baseline NDMI scores, $P = 0.008$ (Supplementary Fig. 4). Because low neutrophil proportions may affect NDMI performance, we simulated the effects of different neutrophil proportions on the probability of detecting DEX exposures (Supplementary Fig. 5). Variation of immune proportions among samples and the sample size of exposed and non-exposed subjects were found to be essential factors affecting NDMI performance. Critical values of less than ~20–30% neutrophils could influence the performance of the NDMI as an indicator of DEX exposure. Among 457 glioma subjects in our study, only 2.3% had <40% neutrophil proportions.

## DEX treatments modify the regression of NDMI scores on neutrophil proportions

A fundamental premise of the CellDMC program is that differential methylation occurring within neutrophils will alter the regression of methylation on neutrophil proportions in DEX exposed subjects. We estimated regression parameters for NDMI and neutrophil proportions

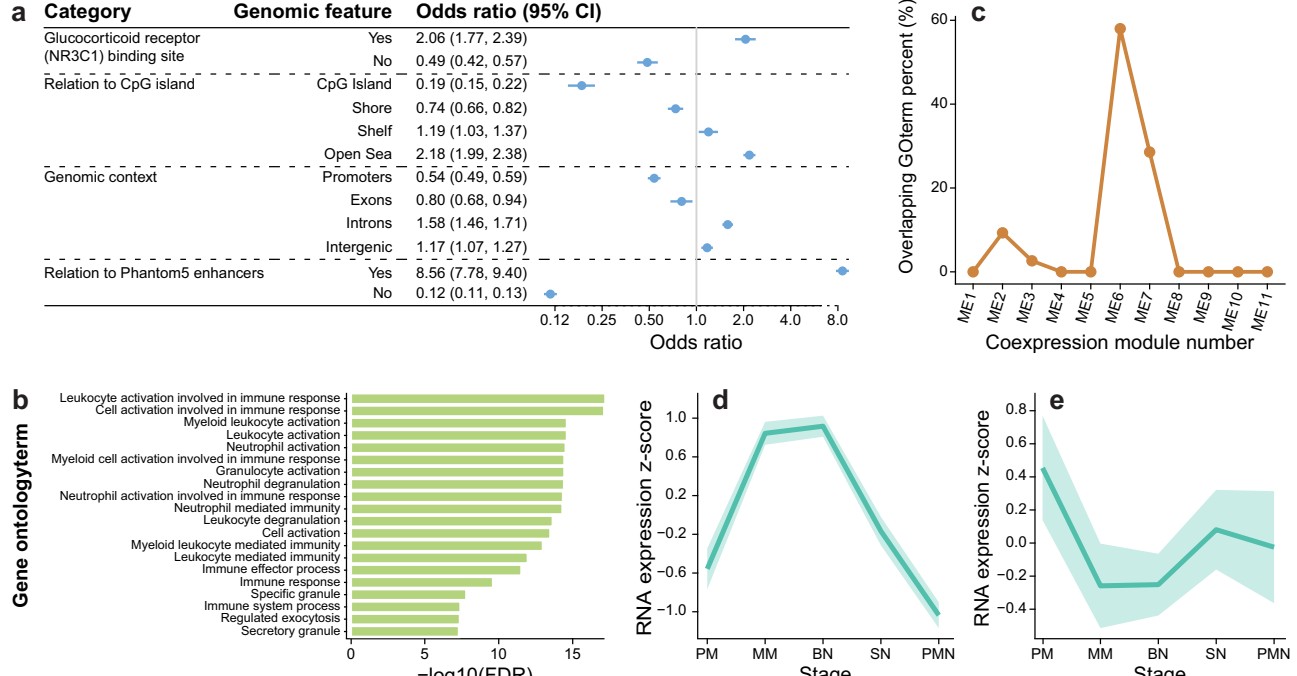

**Fig. 3 | Functional features and gene ontology analyses of genes containing neutrophil-specific DEX loci. a** Genomic context enrichment of neutrophil-specific CpGs ($n = 2621$) with all tested CpGs ($n = 830,277$) as background. **b** Enriched GO terms associated with 1360 genes displaying differentially methylated DEX loci and interaction with neutrophil proportions. **c** Percent of GO terms in the myeloid expression (ME) modules overlapping with the GO terms enriched for 1360 neutrophil-specific loci associated genes. ME6 displayed the highest overlapping percent (58%) with 29 out of 50 overlapping with neutrophil-specific GO terms. **d** Plot of maximal expression of 21 overlapping differentially methylated

neutrophil genes during myelopoiesis; these genes include *CASP10, ALDDH3B1, GAS7, NLRC4, APMAP, OLFM4, EHD4, S100A8, ACAD8, ANKRD22, CNKSR3, FCER1G, UGP2, CLEC12A, OLR1, LRRC57, MAFG, PAPSS2, CD177, LTBR4, LIMS1.* **e** Plot of maximal expression of 16 genes contained in the NDMI algorithm (see Supplementary Table 4) showing maximal expression in promyelocyte and PMN stages of myelopoiesis. PM promyelocyte, MM metamyelocyte, BN band neutrophil, SN segmented neutrophil, PMN polymorphonuclear neutrophil. Source data are provided as a Source Data file.

in training ($N = 135$) and an extended case/control dataset ($N = 602$). In both, the ANOVA linear regression interaction terms of NDMI and DEX exposure were highly statistically significant ($P < 0.0001$). DEX treatments increased the slope gradient of NDMI regressed on neutrophil proportions among DEX exposed subjects (Fig. 4c, d).

## NDMI is a highly accurate predictor of DEX exposure in an independent dataset

We assessed the quality of our predictive models in an independent set of DEX exposed cases and non-glioma controls (Fig. 5a). Receiver operating characteristic (ROC) analyses in the test population indicated high sensitivity and specificity of the NDMI score as a binary predictor of current DEX exposure (Fig. 5b). ROC performance was highest for the NDMI score (0.91; 95% CI 0.871, 0.95) although leukocyte composition performed well (AUC 0.71; 95% CI 0.649, 0.77). The non-cell-specific methylation predictor appeared superior to the leukocyte composition model (AUC 0.84; 95% CI 0.793, 0.896). We also built a version of the NDMI using only presurgery glioblastoma (GBM) ($N = 71$) subjects. The performance of the GBM only NDMI (AUC 0.88; 95% CI 0.83, 0.92) was greater than the leukocyte composition model but less than the NDMI created using all grades of glioma.

## NDMI scores associated with depressed CD4 T cell counts and increased mMDSC levels

Depressed CD4 T cell counts signal immunosuppression; among the 135 training set subjects, 47 (34.8%) had CD4 counts <500 cells/μl and 16 (11.8%) had counts <200 cells/μl (Table 1). Current DEX use was recorded in 39 (83%) and 15 (94%) of these subjects, respectively. NDMI, total white cell count, and their interaction were significantly associated with CD4 T-cell levels in whole blood and explained

substantial variation in CD4 counts among glioma subjects ($r^2 = 0.64$; $p < 0.001$) (Table 2). CD4 counts were positively associated with total white cells, whereas NDMI scores were inversely associated. In these models, DEX exposure variables were not statistically significant, although daily DEX dose was inversely associated with CD4 counts in univariate tests ($r^2 = 0.21$; $p = 0.004$). A subset of subjects' bloods was run in parallel using flow cytometry (FCM) and methylation deconvolution. Bland-Altman's analysis of these paired data showed DNA methylation deconvolution to be highly accurate in estimating CD4 T cell concentrations (Supplementary Fig 6).

We estimated blood mMDSC levels both as proportions of monocytes and as concentrations in 38 consecutive glioma volunteers using FCM (Fig. 6). Subjects taking DEX had greater proportions ($p = 0.002$) and concentrations ($p = 0.005$) of CD14⁺HLA-DR^neg/low mMDSCs compared to non-exposed subjects. We further subdivided the mMDSC population by CD16 expression and found significant DEX-related increases in CD14⁺HLA-DR^neg/low CD16⁻ cells/μl ($p = 0.005$). NDMI scores were positively correlated with total mMDSCs proportions ($r^2 = 0.30$; $p = 0.003$) and concentrations ($r^2 = 0.33$; $p = 0.001$) as well as CD16⁻ mMDSC populations (proportions $r^2 = 0.31$; $p = 0.002$; concentrations $r^2 = 0.34$; $p < 0.0001$). The most robust model predicting mMDSC concentrations included monocyte counts. NDMI and monocyte counts together were positively associated with mMDSC levels in whole blood and together explained 76% of the variation in mMDSC levels and 77% of the variation of CD16⁻ mMDSCs ($p < 0.001$). Average daily DEX exposure, although univariately associated with mMDSCs ($r^2 = 0.24$; $p = 0.04$), was not significant in models containing monocytes and NDMI. Neither cumulative mg of DEX nor duration of use were significantly associated with variations in mMDSC levels ($P > 0.05$).

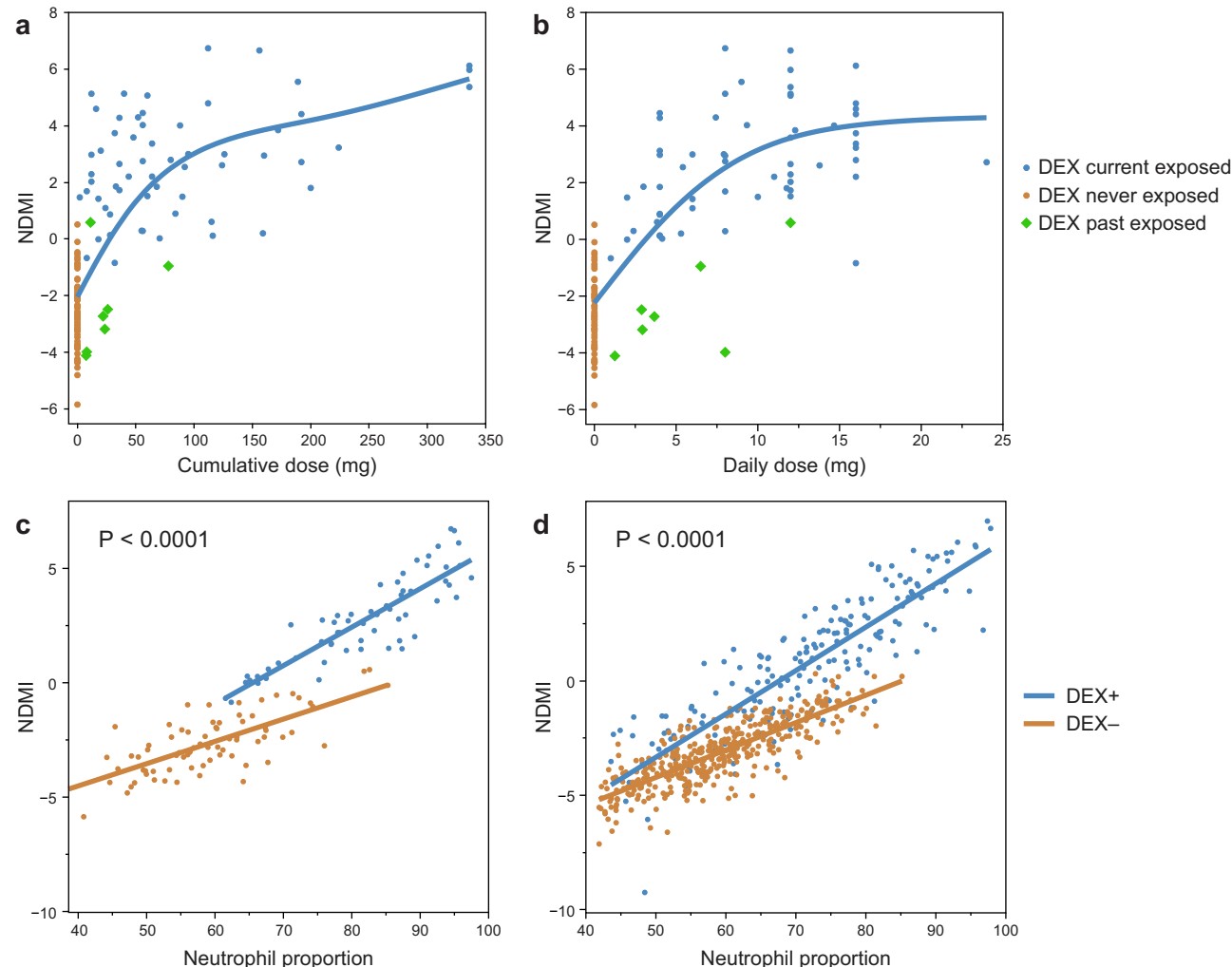

**Fig. 4 | Effects of DEX on NDMI response and modification of NDMI regression on neutrophil proportions. a** Dose–response of cumulative mg of DEX versus blood NDMI scores. **b** Dose–response of average daily mg DEX versus blood NDMI scores. Smooth curves are shown. In **a**, **b**, the 7 past exposed subjects received DEX 4, 7, 8, 10, 17, 18, and 22 days prior to blood draw. **c** Scatterplot of NDMI (y-axis) by the predicted relative fraction of neutrophils (x-axis) between 135 DEX+/DEX− subjects (different colored points) in the training dataset. The lines in the plot reflect the fitted regression line for DEX+ and DEX− subjects. The *p* value resulted from a test of equality of slopes between the DEX+ and DEX− regression lines. **d** Scatterplot of NDMI (y-axis) by the predicted relative fraction of neutrophils (x-axis) between DEX+/DEX− subjects (different colored points) in an independent dataset of 99 exposed glioma subjects and 453 control subjects. The lines in the plot reflect the fitted regression line for DEX+ and DEX− subjects. The *p* value resulted from a test of equality of slopes between the DEX+ and DEX− regression lines. Source data are provided as a Source Data file.

## Elevated NDMI scores in isolated DEX exposed monocytes, mMDSCs and neutrophils

To provide a reference for comparing NDMI scores in different immune subsets we compared DEX exposed cells and 12 different DEX naïve cell subtypes. In DEX naïve cells we observed significantly greater NDMI scores among myeloid-derived cells compared to lymphoid cells; Fig. 6d. In vivo DEX exposed monocytes and mMDSCs contained elevated NDMI scores compared to non-exposed monocytes (*p* = 0.001) and compared to paired DEX exposed monocytes from the same individuals (*p* = 0.03). Monocytes from DEX exposed subjects had higher NDMI scores compared to non-exposed monocytes (*p* = 0.002) (Fig. 6e). To test whether isolated neutrophils exposed in vivo to DEX contain DNA methylation changes consistent with the CellDMC computational method, we isolated cells from 5 DEX exposed glioma patients. The purity of these neutrophil isolates was estimated to be 99.9% (range 99.3–100%) using an extensive methylation deconvolution method[39]. DEX exposed isolated neutrophils exhibited NDMI scores markedly higher than unexposed cells and highest among all isolated cells examined (Fig. 6e). We compared the NDMI scores of isolated neutrophils exposed to DEX to the respective whole-blood NDMI scores from the same individuals; in each of the subjects the neutrophil NDMI score was significantly greater than the whole-blood score (paired *t* test *p* = 0.01; Supplementary Table 10). Further, of the 2621 neutrophil-specific DEX associated CpG sites identified in the CellDMC analysis, 1903 (72.6%) exhibited the same direction of association in isolated cells and 823 probes were nominally statistically significant (*p* < 0.05) between exposed and non-exposed neutrophils. Of the 823 CpG, 683 exhibited the same direction of association (83%) as the CellDMC analysis. Of the 28 CpGs in the NDMI algorithm 9 (32%) exhibited a nominally statistically significant *p* value (*p* < 0.05) and a consistent direction of association as compared with the CellDMC analysis.

## NDMI score is a prognostic marker of glioma survival

To evaluate the NDMI score as a prognostic marker in glioma, we entered patients' NDMI scores into a survival elastic net model that included conventional risk factors of age, gender, tumor classification, tumor location, surgery and chemoradiation treatments, and DEX use

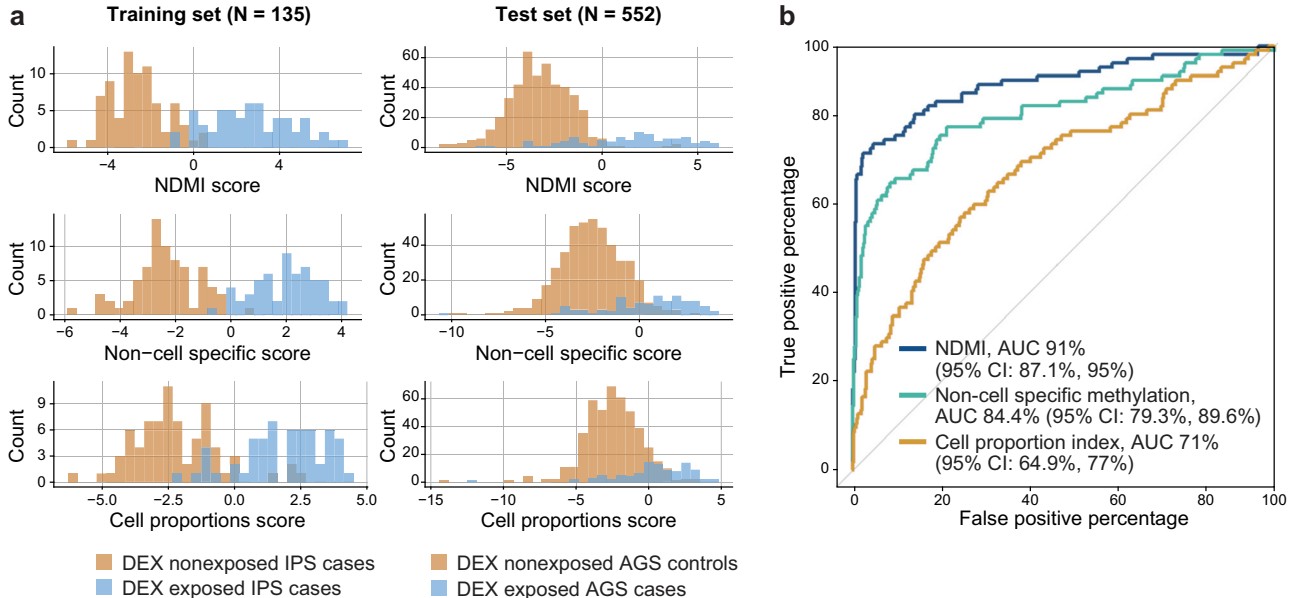

**Fig. 5 | Comparison of DEX predictors in training and test datasets.**
**a** Distributions of NDMI and two other DEX predictor scores in training and test datasets. **b** ROC analysis comparing the performance of three predictors of DEX exposure (NDMI, non-cell-specific methylation, blood leukocyte composition). Source data are provided as a Source Data file.

**Table 1 | Immune parameters for 38 Immune Profiles Study glioma patients and their neutrophil dexamethasone methylation index (NDMI) stratified by dexamethasone use**

| Variables | Dexamethasone at Blood Draw | | | | | | | | P value[a] |
|---|---|---|---|---|---|---|---|---|---|
| | Yes (*n* = 19) | | | | No (*n* = 19) | | | | |
| | Mean | Median | StDev | Range | Mean | Median | StDev | Range | |
| Average daily DEX dose[b] (mg/day) | 8.85 | 8 | 5.21 | (2, 16) | NA | NA | NA | NA | NA |
| Age at Surgery (yrs) | 54.8 | 57 | 17.43 | (21, 79) | 47.84 | 49 | 13.28 | (28, 72) | 0.18 |
| NDMI Score | 2.36 | 2.7 | 1.55 | (0.02, 5.37) | −1.73 | −1.72 | 1.14 | (−3.84, 0.51) | <0.0001 |
| Absolute B cells (cells/μl) | 310.4 | 279.8 | 215.7 | (47.5, 846.1) | 281.4 | 272.2 | 123.9 | (70.2, 545.6) | 0.62 |
| Absolute CD4 (cells/μl) | 517.6 | 440.5 | 443.8 | (46.8, 1744.8) | 930.5 | 848.7 | 466.9 | (275.4, 1825.9) | 0.008 |
| Absolute CD8 (cells/μl) | 374 | 398.3 | 223.1 | (43.6, 818.1) | 468.1 | 447 | 230.1 | (130.1, 1032.2) | 0.21 |
| Absolute Monocytes (cells/μl) | 536.7 | 524.8 | 357.2 | (96.5, 1238.2) | 489.2 | 474.8 | 140.2 | (288.9, 754.6) | 0.59 |
| Absolute NK Cells (cells/μl) | 246.8 | 222 | 129.5 | (114.5, 671.6) | 393.6 | 365.7 | 204.3 | (146.4, 871.3) | 0.01 |
| Absolute Neutrophils (cells/μl) | 8651.2 | 7475.3 | 4527 | (1332.3, 17802) | 4180 | 3335.6 | 2369.1 | (1472.4, 9402) | 0.0007 |
| mMDSC CD16− (cells/μl)[c] | 95.9 | 72.3 | 84.7 | (2.63, 249.9) | 24.5 | 17.2 | 28.8 | (0, 123.5) | 0.002 |
| mMDSC CD16+ (cells/μl)[c] | 28.6 | 15.5 | 31.1 | (0.53, 94.4) | 13.1 | 1.8 | 27.3 | (0, 98.5) | 0.11 |
| mMDSC as a % of Monocytes[c] | 22.1 | 20 | 15.4 | (1.48, 52.2) | 8.7 | 7 | 8.3 | (0, 28.8) | 0.002 |
| mMDSC counts (cells/μl)[c] | 124.4 | 101.2 | 112.1 | (3.16, 339.5) | 37.7 | 22.2 | 55.3 | (0, 221.9) | 0.005 |
| **Categorical variables** | **Number** | **%** | | | **Number** | **%** | | | **P value[a]** |
| **Gender** | | | | | | | | | |
| Male | 14 | 0.74 | | | 14 | 74% | | | 1 |
| Female | 5 | 0.26 | | | 5 | 26% | | | |
| **Tumor Grade** | | | | | | | | | |
| Grade 4 | 17 | 0.895 | | | 7 | 36.80% | | | 0.003 |
| Grade 3 | 1 | 5.30% | | | 4 | 21.10% | | | |
| Grade 2 | 1 | 5.30% | | | 8 | 42.10% | | | |

[a]T-test for continuous variables, chi-squared for categorical variables
[b]Estimated by dividing the cumulative mg of DEX reported by the number of days on DEX
[c]Measured by FCM other immune parameters estimated by methylation deconvolution

at blood draw as well as neutrophil and other immune cell proportions. NDMI score, but not DEX use at blood draw or neutrophil proportion, remained a significant predictor of survival time in a final model that included age, surgery, tumor location, WHO subtype, body mass index, CD4 T cell proportion, and an interaction between WHO subtype and NDMI score (Table 3). Figure 7 shows the Kaplan–Meier survival curves for NDMI score split at the median in the test set's isocitrate dehydrogenase (*IDH*) wildtype GBM and *IDH*-wildtype

**Table 2 | Neutrophil dexamethasone methylation index (NDMI) scores predict CD4 T and mMDSC cell counts in Immune Profiles Study glioma patients with presurgery samples[a]**

| Model 1 (CD4 T cell) | Parameter estimates (dependent variable = absolute CD4 counts) | | | |
|---|---|---|---|---|
| | Predictor | Estimate | Std Error | t Ratio | P value |
| | Intercept | −58.27 | 75.84 | −0.77 | 0.44 |
| | NDMI Score | −158.78 | 11.21 | −14.17 | <0.0001 |
| | Total WBC Count | 0.10 | 0.009 | 11.62 | <0.0001 |
| | Interaction (NDMI*Total WBC) | −0.01 | 0.002 | −5.69 | <0.0001 |
| | **Summary of Fit** | | | | |
| | $R^2$ = 0.64, Adj $R^2$ = 0.63 Observations = 135 F-Ratio = 76.23 ($p$ < 0.001) | | | | |
| **Model 2 (mMDSC)** | Parameter estimates (dependent variable = mMDSCs) | | | |
| | Predictor | Estimate | Std Error | t Ratio | P value |
| | Intercept | −37.78 | 14.30 | −2.64 | 0.013 |
| | NDMI Score | 14.79 | 2.80 | 5.28 | <0.0001 |
| | Absolute_Monocytes | 0.20 | 0.025 | 8.09 | <0.0001 |
| | **Summary of Fit** | | | | |
| | $R^2$ = 0.76, Adj $R^2$ = 0.74 Observations = 36[b] F-Ratio = 52.56 ($p$ < 0.001) | | | | |
| **Model 3 (CD16⁻ mMDSC)** | Parameter estimates (dependent variable = CD16⁻ mMDSCs) | | | |
| | Predictor | Estimate | Std Error | t Ratio | P value |
| | Intercept | −26.88 | 10.50 | −2.56 | 0.015 |
| | NDMI Score | 11.99 | 2.06 | 5.83 | <0.0001 |
| | Absolute_Monocytes | 0.15 | 0.018 | 8.09 | <0.0001 |
| | **Summary of Fit** | | | | |
| | $R^2$ = 0.77, Adj $R^2$ = 0.76 Observations = 36[b] F-Ratio = 56.17 ($p$ < 0.001) | | | | |

[a]Model 1 included all 135 IPS presurgery samples, Models 2 and 3 included 36 presurgery samples with FCM
[b]Two outliers were removed from the model

astrocytoma patients, in each subtype survival times were statistically significantly different. Survival among patients with oligodendrogliomas was not significantly different by NDMI status. In a subset of 74 glioma subjects we evaluated blood NDMI, immune cell estimates and clinical variables in multivariate Cox proportional hazards survival models with extent of surgical resection with and without preoperative tumor volume (Supplementary Tables 11, 12). Importantly, in both models, the NDMI score remained statistically significant as a predictor of survival. In both models, volumetric measurements were significant as were age and WHO tumor classification. DEX use at blood draw was not statistically significant in either model (Supplementary Tables 11, 12).

## Discussion

The present work shows that the DNA methylation-based NDMI measured in whole blood is a robust marker of DEX response that outperforms an approach that is based on the drug's effects on immune cell proportions. Thus, the NDMI could be a potentially useful tool for evaluating glucocorticoid treatments in neuro-oncology. Though further study is needed to demonstrate the full value of this approach, several areas of translational application are anticipated. The NDMI is an objective measure of glucocorticoid response that does not rely on medication logs and associated assumptions about patient adherence. The NDMI appears to capture individual variability in DEX response, as indicated by two measures of immunosuppression (low CD4 and elevated mMDSC counts). The prevalence of both these cell types was best predicted by NDMI scores and not by either daily or cumulative drug doses.

These results provide another datapoint that adds to historical observations[11–13], which have raised concerns about the widespread use of glucocorticoids. Early in diagnostic workup, even the suspicion of GBM can trigger the administration of high-dose DEX in routine clinical practice, regardless of neurological symptoms, tumor size, or extent of cerebral edema. Some of these exposures may be unnecessary and may lead to worsening outcomes. There is a lack of data and consensus in the field concerning optimal glucocorticoid dosing and schedules to guide clinical decision making. An ongoing trial (NCT04266977) is the first of its kind to formally test whether restrictive use of DEX in suspected GBM subjects is an acceptable and safe alternative to current practice, though in symptomatic subjects, DEX may be unavoidable. In these instances, our observation of increased survival risk associated with the NDMI score reinforces the need to titrate to the lowest dose possible[14]. In this regard, the NDMI response may provide a benchmark for calibrating the dose given to an individual patient with the dual goals of providing adequate clinical efficacy and minimizing toxicity, in particular if there is a need to re-start steroids during treatment. Such a precision approach has the potential to significantly reduce steroid-related morbidity in the GBM patient population.

Given the convincing evidence that DEX compromises immunotherapies[14–16] the most important potential translational value of the NDMI score could be as a biomarker of response in immunotherapy trials. As an objective measure of extent of steroid exposure, the NDMI score could be utilized as a stratification factor in such trials to minimize confounding from steroid interference with therapies. Beyond these applications we observed that the NDMI score, but not DEX use, was a significant predictor of survival in *IDH*-wildtype GBM and astrocytoma, though not in oligodendroglioma. Thus, the

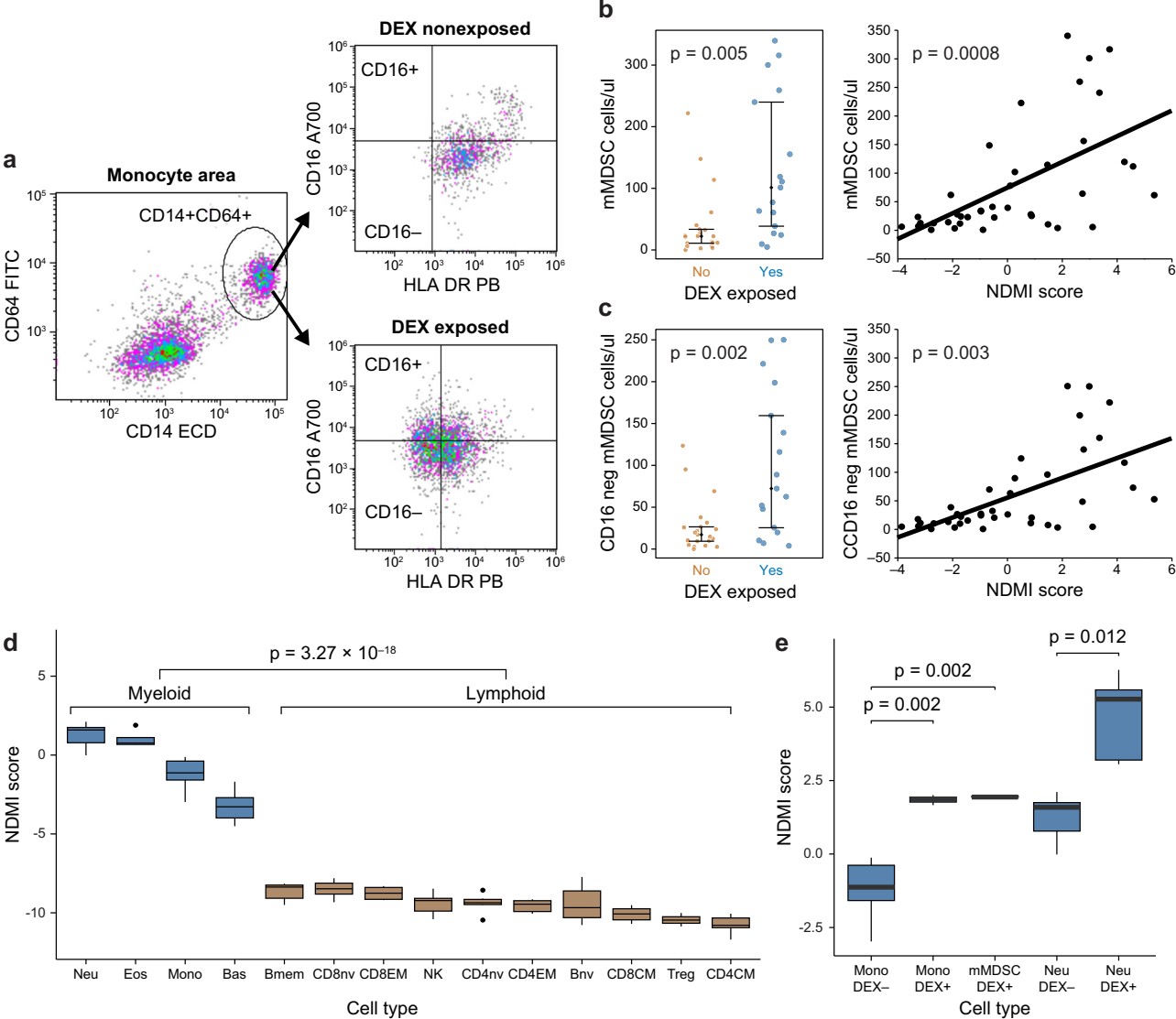

**Fig. 6 | NDMI scores in isolated DEX exposed and non-exposed myeloid and lymphoid blood immune cell subtypes. a** FCM dot plots showing the gating procedure to quantify HLA-DR⁻ CD14⁺ CD16⁻ monocytes (putative mMDSCs). **b** Elevated total mMDSC concentrations in glioma subjects exposed to DEX (n = 38 (19 DEX exposed and 19 non-DEX exposed)) and correlation of total mMDSCs with NDMI scores. **c** Elevated blood concentrations of CD16⁻ mMDSC in glioma subjects exposed to DEX (n = 38 (19 DEX exposed and 19 non-DEX exposed)) and correlation of CD16⁻ mMDSC concentrations with NDMI scores. For **b**, **c**; p values for difference

in mean MDSC values, standard deviations and standard linear regression coefficients were calculated from 2-sided Student's t tests. **d** NDMI scores of isolated non-DEX exposed myeloid cells are statistically significantly higher compared to non-DEX exposed lymphoid cell types. **e** Boxplots showing statistically significant elevations in the NDMI scores of monocytes, mMDSCs and neutrophils isolated by FACS from DEX exposed subjects compared to non-exposed cells. For **d**, **e** median and interquartile ranges are shown, two-sided Wilcoxon test used to calculate p values. Source data are provided as a Source Data file.

NDMI score may be a more relevant prognostic factor in future studies of *IDH*-wildtype GBM.

Given the widespread availability of DNA methylation arrays and the open access Web-based tool that we have created, the NDMI approach is within reach for researchers wishing to assess clinical correlates with this measure. The small volumes of blood samples (0.5 mL) required, and the lack of any labor-intensive special processing needed for DNA methylation-based analyses make it an accessible technique for archival samples, and greatly enhances the value of samples in clinical biobanks. Finally, our observation of elevated NDMI scores among prednisone users in non-glioma controls highlights the applicability of this method to assessment of glucocorticoid use in a range of clinical indications beyond neuro-oncology (e.g., rheumatoid arthritis, lupus, multiple sclerosis, solid organ transplantation). We also note that, consistent with the low systemic absorption associated

with inhaled glucocorticoids[40], we did not observe altered NDMI scores in controls reporting inhaled glucocorticoid use. This is consistent with a recent study in pediatric asthma[41] that failed to find consistent alterations in DNA methylation among users of inhaled glucocorticoids.

The acute effects of DEX and other glucocorticoids on peripheral blood myeloid and lymphoid cells are well described and reflect the intimate linkage of the glucocorticoid pathway with hematopoiesis and immune regulation[42,43]. We add to this knowledge that blood neutrophils dominate the leukocyte methylomic profile of DEX exposure. This result is reminiscent of earlier studies of glucocorticoid-induced gene transcription showing that the numbers of glucocorticoid-responsive genes were highest in hematopoietic compared to nonhematopoietic cells and that neutrophils had the strongest transcriptional response of all cell types studied[36]. Moreover, a

heightened response to glucocorticoids by neutrophils is striking given their terminally differentiated state and short lifespan as well as their limited transcriptomic capacity compared with other

**Table 3 | Neutrophil dexamethasone methylation index (NDMI) is a prognostic factor in glioma survival: Cox multivariate model of NDMI and glioma survival in 429 UCSF Adult Glioma Study patients**

| Variable | Hazard ratio | 95% CI | P value** |
|---|---|---|---|
| NDMI score | 1.43 | 1.29–1.59 | $8.5 \times 10^{-12}$ |
| Age (yrs, continuous) | 1.04 | 1.03–1.05 | $2 \times 10^{-16}$ |
| **World Health Organization 2016 classification** | | | |
| IDH-WT GBM | 1.89 | 1.44–2.51 | $6.8 \times 10^{-6}$ |
| IDH-MT astrocytoma | 0.33 | 0.06–1.95 | 0.2 |
| Oligodendroglioma | 0.18 | 0.10–0.33 | $2.3 \times 10^{-8}$ |
| IDH-WT astrocytoma (baseline) | 1.0 | NA | NA |
| **Interaction of NDMI score with World Health Organization 2016 classification** | | | |
| IDH-WT GBM | 0.85 | 0.76–0.94 | $2 \times 10^{-3}$ |
| IDH-MT astrocytoma | 0.56 | 0.31–1.03 | 0.06 |
| Oligodendroglioma | 0.74 | 0.59–0.92 | $7.7 \times 10^{-3}$ |
| IDH-WT astrocytoma (baseline[a]) | 1.0 | NA | NA |
| Surgery (biopsy versus resection) | 0.52 | 0.37–0.73 | $1.5 \times 10^{-4}$ |
| CD4 T cells (proportions) | 1.04 | 1.01–1.07 | $2 \times 10^{-3}$ |
| BMI (basal metabolic index) | 1.05 | 1.02–1.07 | $1.2 \times 10^{-4}$ |
| Race (white vs. non-white) | 1.47 | 1.07–2.03 | 0.02 |
| **Tumor location** | | | |
| Occipital lobe | 3.34 | 1.73–6.47 | $3.4 \times 10^{-4}$ |
| Overlapping sites | 1.56 | 1.14–2.14 | $5.8 \times 10^{-3}$ |
| Parietal lobe | 0.65 | 0.45–0.95 | 0.03 |
| Frontal/cerebrum/temporal/other (baseline) | 1.0 | NA | NA |

**P values were calculated using the Wald Test

[a]IDH-WT Astrocytoma chosen as comparator baseline for the model

leukocytes[44]. We found that DEX-responsive methylation in neutrophils was enriched in distal regulatory elements and glucocorticoid receptor binding sites. Consistent with previous work, we also showed that DEX-related differential methylation occurs at preexisting accessible chromatin locations as evidenced by their partially demethylated status in unexposed cells and occurring in a lineage-specific fashion. Among lymphocytes, the same sites were typically densely methylated. A robust association of differentially methylated genes encoding granule proteins and neutrophil activation functions was found in gene ontology analysis. Twenty-one of the neutrophil enriched genes overlapped with a previously described co-expression module in human bone marrow whose peak transcriptional expression occurs during the metamyelocyte and band neutrophil stages of myelopoiesis. In contrast, the group of 16 genes that comprise the NDMI predictor were distinct in that they did not belong to the same co-expression group and instead showed peak expression in the promyelocitic and PMN stages of myelopoiesis. Taken together, these results indicate functional heterogeneity among genes targeted by DEX differential methylation.

Isolated neutrophils from DEX exposed subjects displayed the highest NDMI scores of all isolated cells and in each case the scores were elevated above paired whole-blood values from the same donors. These observations are consistent with our bioinformatic predictions. However, despite the lack of statistically significant interactions of methylation values and monocyte proportions, our methylation and FCM studies of mMDSCs also point to DEX-mediated epigenetic modification in subpopulations of CD14[+] monocytes. We confirmed previously reported increased levels of mMDSCs among current DEX users and further those variations in mMDSCs were significantly correlated with individual NDMI scores. We also provide evidence that the monocyte count in conjunction with the NDMI score may improve our understanding of individual variations in mMDSCs. CD16[−] mMDSCs also demonstrated consistent associations with DEX exposure. Earlier studies reported that CD16[−] mMDSC populations display superior immune-suppressive activity compared with CD16[+] fractions[45]. Significantly, of the three DEX exposure

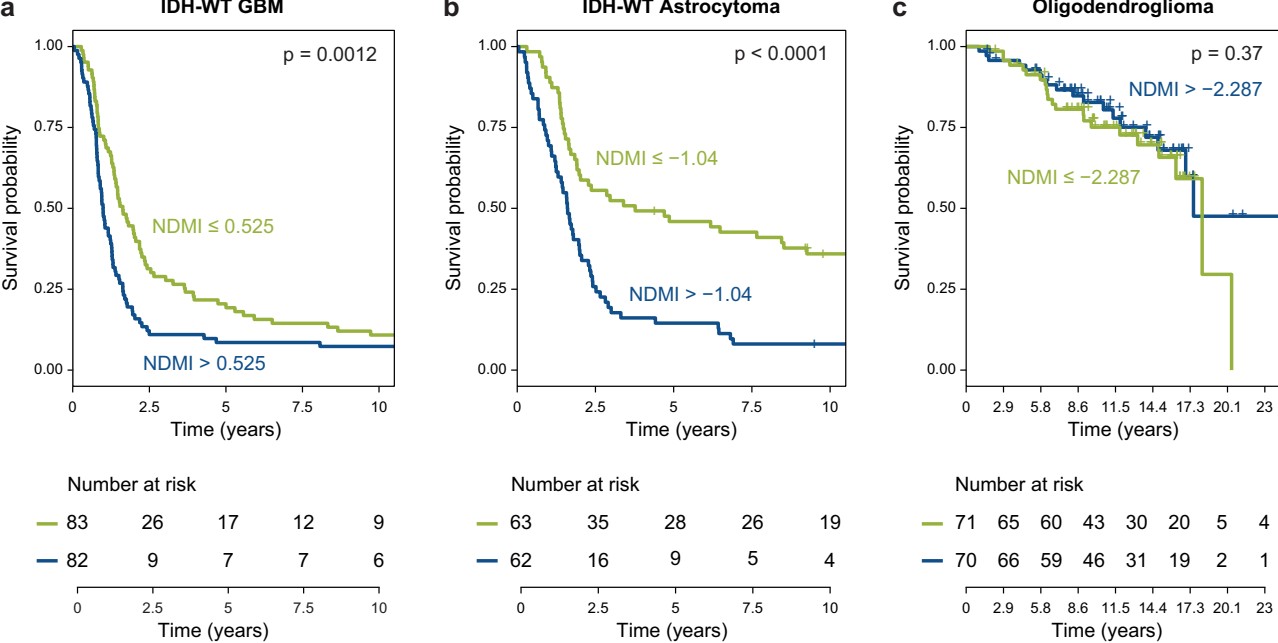

**Fig. 7 | Association of NDMI scores and glioma survival. a** The figure shows the Kaplan–Meier survival curves for NDMI score split at the median in the test set's *IDH*-wildtype GBM patients. **b** The figure shows the Kaplan–Meier survival curves for NDMI score split at the median in the test set's *IDH*-wildtype Astrocytoma patients. **c** Survival among patients with oligodendrogliomas was not significantly different by NDMI status. Source data are provided as a Source Data file.

measures, only the average daily dose was associated with mMDSC concentrations. In models that contained monocytes and NDMI, daily DEX dose became insignificant, suggesting that the NDMI score captures individual variation in the epigenetic response to DEX treatments. Furthermore, the NDMI scores of isolated mMDSC and paired CD14$^+$ HLA-DR$^+$ monocytes from DEX exposed subjects were elevated compared to non-exposed monocytes. Based on these observations, we hypothesize that DEX influences both the neutrophil and monocyte cell methylomes, perhaps reflecting the shared DNA methylation landscape of DEX-responsive regions in both lineages. We have confirmed previously reported deficits in circulating CD4 T cell levels among glioma subjects prior to chemoradiation; 11.8% of our training population demonstrated <200 CD4 T cells/µl. A model incorporating an interaction between the NDMI, and total white cell count provided the strongest predictor of variations in CD4 levels. DEX exposure or clinicopathological variables did not improve model performance. Thus, although excess mMDSCs and low CD4 T cells often occur together in cancer patients, our data suggest that different features of the immune profile in conjunction with the NDMI can be useful to describe variation in these two important immunological parameters.

The current work has several limitations. First, our survival studies were not a randomized controlled study and in the biomarker development phase detailed longitudinal studies were not available to estimate the kinetics of change in DNA methylation during initiation of drug therapy or the recovery time following cessation. Our anecdotical observations indicate that NDMI score can return to pre-therapy levels following ~7 days. Second, while the empirical associations of DNA methylation with DEX treatments are robust, a mechanistic understanding of these effects is lacking. At the tissue level, it is unknown whether DEX-related changes in DNA methylation arise through redistribution of preexisting subpopulations of cells in the bone marrow or marginated pools or instead reflect CpG level modification within stem/progenitor cells during myelopoiesis. At the molecular level, glucocorticoid ligand-triggered demethylation of single CpGs at de novo sites are rapid and replication independent, suggesting an active enzymatic mechanism[32]. In rodent embryonic neural stem cells, Tet3, a member of the ten-eleven translocation family of methylcytosine dioxygenases, was implicated in mediating the effects of DEX on CpG methylation[33]. Whether similar mechanisms apply in adult human myelopoiesis is unknown. Finally, our studies have not assessed gene expression in DEX exposed cells. Recent studies in severe COVID-19 have shown that in vivo DEX exposures induce dramatic changes in peripheral blood neutrophil states leading to immunosuppressive cell subtypes not observed in unexposed subjects[37]. The relationship of different transcriptional states with DNA methylation could provide a mechanistic link between DEX exposures, immune profiles, and glioma survival. In conclusion, we find that alterations in DNA methylation of peripheral whole blood are sensitive indicators of DEX response and are potential pharmacodynamic markers of individual sensitivity to epigenetic modification that are bridged to glioma survival, elevated immunosuppressive myeloid cells and depletion of critical T cell populations.

## Methods

### Patient and control samples

These glioma studies were approved by the Institutional Review Board of the University of California, San Francisco, Human Research Protection Program in the Office of Ethics and Compliance under UCSF Federal wide Assurance 00000068 and met all relevant ethical regulations. Informed consent was obtained from all study participants. The study design utilized a training dataset to develop predictors of DEX exposure and a test dataset to evaluate their performance Fig. 1a. The training dataset included 135 glioma subjects (59 exposed to DEX,

76 not exposed to DEX) sampled before surgery, radiation, or temozolomide treatments. To discover DEX-responsive CpG sites, we carried out an EWAS and then applied those results to the CellDMC program, a method to detect DNA methylation within a specific cell type, Fig. 1b. CellDMC is a linear modeling program that tests each locus associated with DEX to interact with the relative abundance of 6 leukocyte cell types.[46] Separate models were evaluated for each CpG probe and its interaction with CD4 T, CD8 T, B-cell, NK, monocyte, or neutrophil proportions. The training dataset was drawn from the UCSF Immune Profiles Study (IPS), a prospective neuro-oncology clinic-based collection of blood samples, imaging, and other clinical data from adult glioma patients. Presurgery blood samples were typically taken the day before surgery; none were obtained during or after exposure to anesthesia. Blood samples were transferred the same day as drawn for determining total cell counts of specific immune cell types, further processing, and FCM study. At time of blood draws, we also collected a presurgery questionnaire to document daily/cumulative DEX exposure. Additional post-surgery blood samples were collected on some subjects to explore changes in immune parameters following resection. The test dataset ($n = 99$ glioma cases, $n = 454$ controls) and the survival evaluation set ($n = 429$ glioma cases) were drawn from the San Francisco Adult Glioma Study (AGS). AGS was a case-control study of glioma patients newly diagnosed between 1991 and 2012 who were residents of the San Francisco Bay Area or patients of the UCSF Neuro-oncology clinic and age and gender-matched controls without any history of glioma recruited through random digit dialing or from the UCSF blood draw lab[47]. Medical history and current medications were available for all the controls. Patients were selected who had both archival blood and tumor marker data[47]. AGS participants represent newly diagnosed glioma patients; no recurrent or secondary GBM cases are included. In both IPS and AGS cohorts, we classified cases according to the WHO 2016 classification scheme that delineates *IDH*-mutant and wildtype grade IV GBM, grades II and III astrocytoma, and *IDH*-mutant 1p/19q co-deleted oligodendrogliomas. In the AGS, blood samples were collected from patients a median of 100 days after they were histologically diagnosed. Clinical information was collected on patient treatments, including temozolomide chemotherapy, radiation therapy, the extent of surgery, and steroid use at the time of blood sampling. Information on current medication use at blood draw (including any steroid use and names of the types of steroid drugs taken) was obtained from an in-person blood draw questionnaire which collected information on treatments and medications taken at the time of blood draw; medication logs for DEX dosages were not available.

### DNA methylation array

Frozen (−80 °C) anticoagulated whole blood or isolated cells were processed, DNA isolated, and bisulfite converted[48]. All samples and array experiments were performed blinded to clinicopathologic variables. Approximately 200–500 ng of DNA was applied to Illumina DNA methylation EPIC 850K beach chip arrays. Preprocessing and quality control of fluorescence data were accomplished using the minfi Bioconductor package[49]. CpG loci having a sizable fraction (>25%) of detection $p$ values above a predetermined threshold were filtered (detection $P > 10E^{-5}$)[50]. A "noob" background correction method was used to account for background fluorescence and dye bias[51]. Beta-mixture quantile normalization (BMIQ) was performed to correct probe design bias[52]. The presence of technical sources of variability induced by plate or BeadChip was examined using principal components analysis (PCA), the top K principal components[53] were examined in terms of their association with plate and BeadChip. If plate or BeadChip were found to be significantly associated with any of the top K principal components, we applied ComBat method[54] for normalization using the sva Bioconductor package. The final dataset contained 830,277 probes.

## Immunomethylomic assay

Using the preprocessed and normalized methylation data, we applied an optimized reference-based cell mixture deconvolution[55]. This immunomethylomic method provides the proportions of 6 major cell types (CD4 T, CD8 T, B-cell, NK, monocytes, neutrophils) and the neutrophil/lymphocyte, CD4/CD8, and lymphocyte/monocyte ratios. DNA methylation-based immune profiles are highly accurate and reproducible[48,55,56] (Supplementary Table 7, Supplementary Fig. 6). The cell fractions were multiplied by the sample total white cell count when these data were available (IPS samples) to estimate absolute cell counts. Historical AGS cases did not have concurrent cell count data. To characterize the purity of isolated neutrophils and explore the NDMI scores within different immune cells, we employed an expanded deconvolution technique that captures 12 different cell types[39].

## Elastic net regression and creation of three DEX exposure predictors

We built three predictors of DEX exposure using elastic net regression procedures on the following data: (i) blood leukocyte composition, (ii) neutrophil-specific methylation, and iii) non-cell-specific methylation. Developing these markers required blood leukocyte composition, which we derived from a validated DNA methylation deconvolution method[48,55,56]. We then created a leukocyte composition index as input for DEX exposure prediction (i) we utilized 6 cell proportions and ratios of neutrophil/lymphocyte, CD4/CD8 T cell, and lymphocyte/monocytes. To discover in vivo DEX-responsive CpGs in the training dataset for predictors (ii) and (iii), we performed an epigenome-wide association study (EWAS) using the "lmfit" function in the limma package[57] comparing methylation beta values for patients on DEX at the time of blood draw and patients not taking DEX at the time of blood draw. The EWAS yielded a list of probes nominally significantly differentially methylated $p < 0.05$ with an absolute delta beta value ($\geq 0.10$). CpGs identified in EWAS were further analyzed the CellDMC[46] function of the EpiDISH package using $R$ with the default threshold parameters (FDR $\leq 0.05$)[58]. Neutrophil-specific CpG loci ($N = 2621$) and non-cell-specific CpG loci ($N = 17{,}733$) were identified and used to develop the methylation DEX-responsive predictors. Elastic net regression was used to build the DEX exposure classifiers using (i) leukocyte composition, the (ii) neutrophil-specific, and (iii) non-cell-specific CpGs identified from CellDMC[46]. The R glmnet was used to build the elastic net models and cross-validation to identify the optimum lambda value[59]. Each of the three predictors was tested using an external dataset consisting of 99 DEX$^+$ glioma samples and 453 DEX$^-$ control samples. An ROC curve was built to compare the predictive capability of the three models on the same test set[60].

## Genomic features of DEX-responsive CpG methylation

Odds ratio and $p$ values are calculated to test for *NR3C1* binding site enrichment in differentially methylated neutrophil-specific CpGs ($n = 2621$) using the total measured as background (EWAS $n = 830{,}277$). Genomic information for transcription factor NR3C1 binding site is extracted from the ENCODE database "Transcription Factor ChIP-seq Clusters", version wgEncodeRegTfbsClusteredV3. The Illumina HumanMethylationEPIC annotation file and the UCSC Genome Browser UCSC_hg19_refGene file were used to test neutrophil-specific CpGs ($n = 2621$) for genomic context enrichment among all tested CpGs ($n = 830{,}277$). The relation to CpG islands and enhancers was identified from the Illumina HumanMethylationEPIC annotation file for each probe. To define the genomic regions as promoters, introns, exons, or intergenic for each probe, the annotateWithGeneParts function from the R-package genomation and the UCSC_hg19_refGene file were used to map the regions to all CpG loci on the Illumina HumanMethylationEPIC array. If a probe mapped to more than a single genomic region, the probe was assigned preferentially with the order promoters, exons, introns, and intergenic. Fisher's exact tests of $2 \times 2$ tables were conducted to calculate odds ratios (ORs), $p$ values, and 95% confidence interval for CpG island, enhancers, and genomic regions enrichment of neutrophil-specific CpGs compared to all other DEX differentially methylated loci. eFORGE 2.0 was used to assign histone chromatin marks to differentially methylated probes[61] in the 28 CpG NDMI.

## FCM detection of blood mMDSC and CD4 T cells

MDSCs are a heterogeneous population of immature myeloid cells, and the definition of these subsets has not yet reached a consensus. We defined mMDSC immunophenotype consistent with commonly employed markers[62,63], substituting the myeloid marker CD64 for CD33. Relative and absolute quantities of total (CD14$^+$CD64$^+$CD11b$^+$HLA-DR$^{neg/low}$) and CD16 negative (CD14$^+$CD64$^+$CD11b$^+$HLA-DR$^{neg/low}$CD16$^-$) mMDSC populations were estimated. FCM was performed by the UCSF clinical immunology core using established clinical protocols and analytic-specific reagents. Blood samples (3 ml EDTA preserved whole blood) were collected, and total white blood cell (WBC) counts were recorded. Following red blood cell lysis, cells were mixed with a monoclonal antibody mixture including CD45, CD64, CD16, CD11b, HLA-DR (Supplementary Table 2) in a single tube, incubated in the dark for 20 min, washed, and fixed with 0.1% formaldehyde prior to acquisition on the Navios EX Flow Cytometer (Beckman Coulter Life Sciences). Data were analyzed using Kaluza Analysis Software 2.1 (Beckman Coulter Life Sciences). Forward and side scatter were employed to discriminate the non-doublet, non-debris, CD45$^+$ leukocyte population and cells counted (Supplementary Fig. 8). Further CD64$^+$CD14$^+$ gating identified the monocyte population. The HLA-DR$^{neg/low}$ gate in the monocyte population was set in each sample according to the HLA-DR$^{neg/low}$ granulocyte population in the same tube. These HLA-DR$^{neg/low}$ monocytes were further gated for CD16$^{+/-}$ expression. The proportion of each cell type present in the sample was then calculated based on the number of CD45$^+$ leukocytes. The mMDSC proportion of CD45$^+$ leukocytes counted in the acquisition tube was calculated. Total WBC counts were then used to calculate the final cell/µl concentrations of mMDSCs in the blood. Antibodies were used to stain CD45, CD3, and CD4 to identify the CD4 T cells. Within the CD45 population, the low side scatter intensity CD3$^+$ cluster identified the CD3$^+$ population. These CD3$^+$ cells were further gated for CD4$^{+/-}$ expression. The CD4 T-cell proportion of CD45+ leukocytes counted in the acquisition tube was calculated. Total WBC counts were then used to calculate the final cell/µl concentrations of CD4 T cells in the blood. Linear regression models were constructed with NDMI and other factors as predictors to explore CD4 T cell and mMDSC parameters as outcomes. Examination of residuals led to dropping two outliers from mMDSC models; these subjects were of advanced age (i.e., 72, 75 yrs.) and were sampled soon after receiving high DEX doses (i.e., 16 mg/day).

## FACS isolation of mMDSCs, monocytes and neutrophils

Fresh anticoagulated blood was processed within 24 h. For mMDSCs and monocytes: blood mononuclear cells were isolated with 1.077 Histopaque gradients, stained with a cocktail of fluorescently labeled antibodies (CD3, CD56, CD19, CD14, CD11b, CD16, HLA-DR, CD33, and CD15 (Supplementary Table 6) treated with PE/Cyanine7 Streptavidin and resuspended at 1:5000 dilution of SYTOX™ Green. Cells were then run directly on a BD FACSAria™ Fusion cell sorter. Forward scatter hi CD3$^-$, CD19$^-$, CD11b$^+$, CD33$^+$, CD14$^+$, and CD15$^-$ cells were gated and plotted for HLA-DR expression. CD3 HLA-DR$^-$ neg cells and CD19 HLA-DR$^+$ positive B cells were used to set the sorting gate for mMDSC cells lacking HLA-DR expression (i.e., HLA-DR$^{neg/low}$). HLA-DR$^{high}$ cells (normal monocytes) were collected from the same individuals. For neutrophil isolations, a Histopaque two-step gradient (1.119 layered with 1.077) was

used. Granulocytes were collected on the 1.119 layer above the red blood cells and below monocytes and lymphocytes; residual red cells were lysed with ammonium chloride. Cells were FcBlocked and stained with CD3, CD56, CD123, and CD49d in a negative depletion channel. Neutrophils were then sorted as CD19⁻, CD33⁻, and CD66b⁺ cells (Supplementary Table 13). Gated cells were plotted against FSC and SSC to select cells with high forward and side scatter characteristic of granulocytes. All isolated cell pellets were stored at −80 °C until DNA methylation analysis. The purity of isolates was checked using a high-definition immune cell methylation deconvolution method[39].

## Survival analysis

Overall survival is defined as the time from the initial diagnosis to death or last follow-up. Those patients without a date of death were censored at the date of the last follow-up. Multivariate survival models were initially built via regularized cox regression[55] using the *glmnet* package in *R*. Cross-validation was performed to identify the optimum lambda value[50]. Subsequently, forward/backward stepwise feature selection was employed, and interactions were investigated. Covariates with *p* values < 0.05 were retained. For each of the three most common WHO 2016 subtypes (*IDH*-wildtype GBM, *IDH*-wildtype astrocytoma, and oligodendroglioma), the 1st and 3rd quartile of NDMI scores were estimated, and plots of the corresponding Kaplan–Meier curves were drawn.

## Simulation study of NDMI performance and neutrophil proportions

We conducted a series of simulation studies to assess the statistical power for detecting a difference in the mean value of the NDMI between DEX exposed and non-exposed individuals (Supplementary Fig. 7). Specifically, our interest centered on determining the nature by which varying fractions of neutrophils in the whole-blood samples used as the basis for calculating NDMI impacts the statistical power for detecting a difference in the mean value of the NDMI between DEX exposed and non-exposed individuals. To achieve this goal, we simulated whole-blood DNA methylation data for DEX exposed and non-exposed individuals, varying the fraction of neutrophils across the study samples from 20% to 80% in increments of 5%. For each simulated dataset, the NDMI was calculated for each sample and used to test the null hypothesis of no difference in the mean NDMI between DEX exposed and non-exposed individuals. For a fixed fraction of neutrophils across the study samples (e.g., 20%), the above process was repeated one-thousand times, and statistical power was calculated as the proportion of times the null hypothesis was correctly rejected.

To simulate in-silico whole-blood methylation data for DEX exposed and non-exposed individuals, we first used a publicly available dataset consisting of cell-specific DNA methylation data profiled using the Illumina HumanMethylationEPIC array for various isolated leukocyte subtypes, including: CD4 T cells, CD8T cells, B cells, natural killer cells, monocytes, and neutrophils[55] GEO: GSE110554) to estimate the cell-specific means and standard deviations of the 28 CpGs that comprise the NDMI. The cell-specific means and standard deviations for the 28 NDMI CpGs were then used to calculate the two shape parameters of the beta-distribution using methods of moments estimation. Using the beta-distribution shape parameters, we next generated cell-specific methylation data, $\mathbf{X}^{(0)} = \left( \mathbf{X}_1^{(0)}, \mathbf{X}_2^{(0)}, \ldots, \mathbf{X}_{n_0}^{(0)} \right)$, for $n_0$ non-exposed individuals by randomly sampling methylation beta values for the 28 NDMI CpGs using the previously estimated shape parameters. Here, $\mathbf{X}_j^{(0)}$ is the cell-specific methylation data for individual $j$, and is a 28 × 6 matrix whose rows represent the 28 NDMI CpGs and whose columns are the 6 leukocyte subtypes. The same process was performed for generating cell-specific methylation for $n_1$ DEX exposed individuals, $\mathbf{X}^{(1)} = \left( \mathbf{X}_1^{(1)}, \mathbf{X}_2^{(1)}, \ldots, \mathbf{X}_{n_1}^{(1)} \right)$, except that the beta values simulated for neutrophils across the 28 NDMI CpGs were generated by imposing a difference in mean methylation consistent with that calculated in a separate dataset consisting of monocyte-specific methylation data in $n = 6$ non-exposed individuals and $n = 3$ DEX exposed individuals. Using the cell-specific methylation data for the $n_0$ non-exposed and $n_1$ DEX exposed individuals, we next generated cell proportions for each of the samples $\mathbf{W} = (\mathbf{w}_1, \mathbf{w}_2, \ldots, \mathbf{w}_N)$ by simulating data from a Dirichlet distribution with total concentration parameter ($\alpha_0$) equal to 18, 73, and 127. Here, $\mathbf{w}_j$ is a 1 × 6 vector representing the cell proportions for subject $j$ and $N = n_0 + n_1$ is the total number of samples, which ranged from $N = 20$ to $N = 80$ in increments of 20. The total concentration parameter is related to the variability in the cellular landscape across samples wherein larger values are associated with less variability in the cellular landscape across samples and conversely, smaller values, more variability in the cellular landscape across samples (Supplementary Fig. 7). The rationale for the choice of the total concentration parameters is described elsewhere[64]. In simulating cell proportions for each of the samples, the Dirichlet parameter for neutrophils was varied such that the mean fraction of neutrophils ranged from 0.20 to 0.80 in increments of 0.05. Having generated the cell-specific methylation data ($\mathbf{X}^{(0)}, \mathbf{X}^{(1)}$) and cell proportions $\mathbf{W}$, we next computed whole-blood methylation signatures for each of the samples $\mathbf{Y} = (\mathbf{Y}_1, \mathbf{Y}_2, \ldots, \mathbf{Y}_N)$ by taking the matrix product of the cell-specific methylation data for a given sample and their simulated cell proportion:

$$\text{non} - \text{exposed} : \mathbf{Y}_j = \mathbf{X}_j^{(0)} \mathbf{w}_j^T \quad (3)$$

$$\text{Dex} - \text{exposed} : \mathbf{Y}_k = \mathbf{X}_k^{(1)} \mathbf{w}_k^T \quad (4)$$

Where $j = 1, 2, \ldots, n_0$ and $k = 1, 2, \ldots, n_1$. Using the whole-blood methylation signatures, $\mathbf{Y} = (\mathbf{Y}_1, \mathbf{Y}_2, \ldots, \mathbf{Y}_N)$ we next calculated the NDMI for each of the $N$ samples and performed a two-sample $t$ test to test the null hypothesis of no difference in the mean NDMI between DEX exposed and non-exposed individuals assuming a type 1-error rate of 5%. For a fixed proportion of neutrophils in whole blood, overall sample size $N$ and the Dirichlet concentration parameter, the process described above was repeated 1000 times and statistical power was calculated as the proportion of times the null hypothesis was correctly rejected.

## Reporting summary

Further information on research design is available in the Nature Research Reporting Summary linked to this article.

## Data availability

Methylation and phenotype data used in this paper are available through dbGaP controlled access. Methylation and phenotype data from the Adult Glioma Study are available through dbGaP Study Accession phs001497.v2.p1. Methylation and phenotype data from the Immune Profiles Study are available through dbGaP Study Accession phs002998.v1.p1. Please note that we do not have IRB approval to release individual level data for 15 out of 457 AGS controls (from Series 1), so these controls are not included in the source files. The remaining data are available within the Article, Supplementary Information. Source data are provided with this paper.

## Code availability

The online application we developed using the R Shiny application to calculate neutrophil dexamethasone methylation index (NDMI) scores can be found at the following URL: https://btcdb.shinyapps.io/ndmi/

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

## Acknowledgements

Work at University of California, San Francisco was supported by the National Institutes of Health (grant numbers R01CA52689 (M.W., J.K.W.), P50CA097257 (J.K.W., A.M.M., J.C., M.W.), R01CA126831 (J.K.W.), R01CA139020 (M.W., J.K.W.), and R01CA207360 (J.K.W.), the loglio Collective, the National Brain Tumor Foundation, and by donations from families and friends of John Berardi, Helen Glaser, Elvera Olsen, Raymond E. Cooper, and William Martinusen. J.K.W. is supported by the Robert Magnin Newman Endowed Chair in Neuro-oncology, MW by the Stanley D. Lewis and Virginia S. Lewis Endowment. UCSF Parnassus Flow Core RRID:SCR_018206 and DRC Center Grant NIH P30 DK063720 (S.J.T.). The authors wish to acknowledge study participants, the clinicians, and research staff at the participating medical centers, the UCSF Helen Diller Family Comprehensive Cancer Center Genome Analysis Core, which is supported by a National Cancer Institute Cancer Center Support Grant (5P30CA082103), the UCSF Cancer Registry (for updating UCSF glioma case survival and vital status), and the UCSF Neurosurgery Tissue Bank. This publication was supported by the National Center for Research Resources and the National Center for Advancing Translational Sciences, National Institutes of Health, through UCSF-CTSI Grant Number UL1 RR024131. Its contents are solely the responsibility of the authors and do not necessarily represent the official views of the NIH. The collection of cancer incidence data used in this study was supported by the California Department of Public Health pursuant to California Health and Safety Code Section 103885; Centers for Disease Control and Prevention's (CDC) National Program of Cancer Registries, under cooperative agreement 5NU58DP006344; the National Cancer Institute's Surveillance, Epidemiology and End Results Program under contract HHSN261201800032I awarded to the UCSF, contract HHSN261201800015I awarded to the University of Southern California, and contract HHSN261201800009I awarded to the Public Health Institute, Cancer Registry of Greater California. The ideas and opinions expressed herein are those of the author(s) and do not necessarily reflect the opinions of the State of California, Department of Public Health, the National Cancer Institute, and the Centers for Disease Control and Prevention or their Contractors and Subcontractors. All analyses, interpretations, and conclusions reached in this manuscript from the mortality data are those of the author(s) and not the State of California Department of Public Health. Study data were collected and managed using REDCap electronic data capture tools hosted at the University of California, San Francisco[65,66]. REDCap (Research Electronic Data Capture) is a secure, web-based software platform designed to support data capture for research studies, providing (1) an intuitive interface for validated data capture; (2) audit trails for tracking data manipulation and export procedures; (3) automated export procedures for seamless data downloads to common statistical packages; and (4) procedures for data integration and interoperability with external sources.

## Author contributions

Conceptualization: J.K.W., A.M.M. Writing original draft: J.K.W. Experimental design: A.M.M., J.K.W., D.C.K., L.A.S., H.H., S.J.T., Z.Z., M.W., K.T.K., J.W.T., J.C. Implementation: A.M.M., G.W., L.M., T.R.,

H.H., E.T., S.J.T., C.M.T., Z.Z., D.C.K., L.A.S., P.B., B.C.C., K.T.K. Analysis and interpretation of the data: J.K.W., A.M.M., G.W., D.C.K., E.N., P.B., H.H., L.A.S., B.C.C., M.W., K.T.K,. E.T. Review and editing: All authors were involved in the writing of the paper and have read and approved the final version.

## Competing interests

J.K.W. and K.T.K. are cofounders of Cellintec, which played no role in the current study. The remaining authors declare no competing interests.
