## [Peer Review File · Nature Communications]

DNA methylation as pharmacodynamic markers of glucocorticoid response and glioma survivalREVIEWER COMMENTS

Reviewer #1 (Remarks to the Author): GBM clinical – translational expert

This is an interesting and potentially useful study that describes a blood-based biomarker that can help identify the effect of glucocorticoids (i.e. dexamethasone) on the peripheral immune system in patients with gliomas, who routinely get this immunosuppressive drug. The authors have characterized the DNA methylation profile associated with glucocorticoid administration on a discovery sample set, and validated this on a separate set of samples. The investigators used computational deconvolution algorithms that lead them to attribute the steroid-related DNA methylation profiles encountered to neutrophils. They found that this DNA methylation profile correlates with the use of corticosteroids, with decrease in CD4 T cells and the accumulation of monocytic myeloid-derived suppressor cells. This biomarker was associated with shorter overall survival in glioma patients, outperforming the dose of steroids at the time of blood-draw as a prognostic factor. Moreover, in a non-glioma population, the methylation-based biomarker was increased in the context of history of prednisone use. Interesting analysis show non-linear steroid dose to methylation profile dynamics, as well as temporal/longitudinal analysis where the DNA methylation signature of steroids remain for approximately 1 week after stopping this medication. Overall this biomarker is an elegant approach to characterize the potential iatrogenic immunosuppression that steroids might elicit on glioma patients, an important confounder when evaluating outcomes of glioma related trials, in particular of immunotherapy studies. This is an interesting and valuable study, yet important issues that are worth addressing prior to publication include:

- 1) The deconvolution approach of DNA methylation profile used to attribute the signature encountered to neutrophils is interesting, and the simulation for proportion of neutrophils on the sample adds to the rigor of the approach, but ultimately this was not validated. Proper validation by an independent, complementary experimental approach would be important.
- 2) To best understand the potential use of this biomarker, comparison of its performance to that of cumulative steroid use, or average use of steroids over weeks to a month, with regards to its prognostic properties in glioma patients is important. In other words, does it make sense to use this sophisticated analysis vs the medication log?
- 3) The use of this biomarker as a prognostic factor and as a measure of a confounder variable on immunotherapy trials and glioma outcome studies would require a user-friendly interphase. The authors should make an effort to make this analysis algorithm, and training dataset available for its independent validation and implementation, and spell out how is this signature best “quantified”.
- 4) The discussion is lengthy, could be streamlined.

Reviewer #2 (Remarks to the Author): DNA methylation – plasma expert

Major comments

- A major rewrite is necessary. Results need clear explanation. Figures are not adequately explained and therefore very difficult to properly interpret. Figure legends are inadequate. Conversely, the methods are very long and overly detailed. The paper also lacks a conclusion.
- It is hard to understand the potential application of the NDMI score in a clinical setting. The manuscript needs a better explanation of the main objective of this score. Several times throughout the manuscript, it is proposed that the NDMI score could be used to understand which patients had been exposed to DEX. The rationale behind having an epigenetic signature for DEX exposure is not clear when surely this knowledge is known already. At other instances, the authors imply that the score could be used to understand the effect of DEX in the different populations of immune cells, but the importance of this or potential application is not clear.

Specific comments

- It is unclear whether the EWAS study was adjusted for multiple testing (manuscript only mentions p

value, not adjusted p value). If this was not adjusted, it would be advisable to repeat analysis, since this could alter the number of significant differentially methylated probes between DEX exposed and non-exposed patients.

- GC is a poor choice of abbreviation; it is hard to interpret whether CpG sequences or synthetic glucocorticoids are being discussed. Glucocorticoid is one word and therefore doesn't need an abbreviation!
- Details on size of training set are missing, line 147-148. Inconsistent details given on line 163: 3,164 total subjects in the AGS, and lines 166-167.
- Figure 1A: immune cell deconvolution, survival evaluation set are not explained elsewhere in the manuscript. Overall very challenging to interpret this figure.
- Figure 1B is not cited in text.
- Lines 324-325 - B-cells not clearly indicated on either Fig 2 B or C.
- Figure 2c, colour coding of myeloid and lymphoid is not clear to graph,
- Figure 3 - incorrect labelling in figure legend (first table not described in legend)
- Figure 3A: For GR binding sites, is the receptor binding to these sites or is DEX?
- Figure 3C ME1-11 are shown on the x-axis, but in text EM6 is described (line 335), possible typo?
- Figure 3E not described in figure legend.
- Prednisone not mentioned in methods, only abstract and results (line 371), data not shown in graphs How and where were these data collected?
- Figure 4, colour difference between current and past DEX is difficult to see. In addition, "past DEX" is not clearly defined. What has been the time difference between DEX dose and recording data? Are subjects here plotted twice for previous dose with DEX and follow up Dose of DEX "current DEX"?
- Figure 4: is this plotting only the training set or training and test? This is unclear, see lines 373-376. This should also be specified in the figure legend.
- Lines 396-397. Is this to day that DEX exposure results in a higher methylation level in neutrophils or does it mean that methylation simply varies more?
- Fig 5A: Non-cell specific score and cell proportion scores need proper explanation
- The text does not specifically reference Fig 6A, B or C. P values should be added to the graphs
- Figure 7 GBM, IDH abbreviations in legend not defined elsewhere in text.
- Lines 315-316: predominant effect of DEX is hypomethylation. Needs more explanation.
- Line 456: Are the pre-surgery subjects glioma patients? Needs clarification.
- Lines 500-502: These are not shown, results do not go through any data on chromatin accessibility and DNA methylation.
- Line 515: Mention that they looked at histone methylation, but no results have been shown where they conducted this analysis .
- Methods (lines 216-217): There is mention of an external dataset with 311 DEX- control samples. It is unclear whether these are also glioma samples.
- Results + Figure 7: Authors should take advantage of the number of patients in the test dataset to do survival analysis. Instead they use hypothetical patients (only a few) to show correlation between NDMI scores and survival prognosis. This might lead to a biased interpretation of the data and its potential use in the clinical setting.

Reviewer #3 (Remarks to the Author): DNA methylation – liquid biopsy expert

The authors present a novel blood based biomarker, coined NDMI, that provides a measure of epigenetic response of subjects to dexamethasone. Dexamethasone is a commonly used steroid with a wide range of side effects and there exist few ways of predicting an individual's response. A few questions that may strengthen the manuscript:

1) The use of dexamethasone in the pre-, peri-, and post-operative management of glioma is an important, and debated, clinical question. While NDMI may be a prognostic factor, it remains unclear to me how this would change the care of patients with glioma. If a neuro-oncologist is provided with a patient's NDMI score, what would they do differently for a given patient? If a patient is symptomatic with significant vasogenic edema and an NDMI score that suggests poor dexamethasone response,

there are few alternatives.

2) What is the performance of the NDMI score if built and validated on GBM samples alone, the patients in whom prolonged dexamethasone treatment is most likely? The other brain tumor patients are less likely to require prolonged dexamethasone use.

3) Key clinical variables are not accounted for in the survival analysis: tumor volume, tumor location, and extent of resection (biopsy vs. any resection is not ideal).

4) While preclinical data and retrospective studies, as cited by the authors, the association between dexamethasone and poor survival in GBM has not been tested in an RCT. This should be noted.

5) Figure 7A and 7B, the text and legend describe the survival curves for IDH WT GBM and IDH WT astrocytoma, respectively. However, the number at risk tables for both 7A and 7B are the same. Please clarify.

6) Increasing number of studies have shown that many IDH WT astrocytomas have very similar survival to GBM. Why is there not a statistically significant association between survival in IDH WT astrocytomas in Table 2 and Figure 7?

7) The authors comment that there is substantial variability in NDMI scores in glioma patients and this variability is not explained by dexamethasone dose or exposure. This variability is quite notable in Figure 4. They suggest that pre-drug methylation status is an important variable. How do the authors propose accounting for this in clinical practice?

8) What conditions did the control individuals exposed to prednisone have? Would be very interesting if the authors could test other inflammatory conditions in the brain treated with steroids as controls.

Reviewer #4 (Remarks to the Author): glucocorticoid in cancer expert

Dr. Kelsey and colleagues investigated the relevance of glucocorticoid therapies and DNA methylation remodeling in peripheral immune compartment. With tools and strategies including epigenome-wide association, CellDMC, immune cell deconvolution, and elastic net regression, they developed a biomarker for dexamethasone (DEX) response, called neutrophil DEX methylation index (NDMI). By data mining of the UCSF Immune Profiles Study and the San Francisco Adult Glioma Study, used as the training and test cohort respectively, they show that NDMI exhibited prognostic values for the survival of glioblastoma and astrocytomas. Also, high NDMI was found to correlate with depressed CD4 T cells and elevated monocytic MDSC in DEX-treated individuals, indicating an immunocompromised status. Thus, this study proposed NDMI as a highly sensitive and specific epigenetic biomarker for current DEX-exposure with prognostic values of glioma survival. In general, several major concerns should be carefully addressed to consolidate the conclusions.

1. Can major conclusions be extrapolated to other cancer types? Does NDMI only exhibit prognostic values in certain subtypes of glioma?

2. Tumor progression can augment the production of endogenous glucocorticoid and activate GR signaling. Do high levels of endogenous glucocorticoid correlate with increased NDMI in glioma cancer patients? Do different types of synthetic glucocorticoids alter NDMI with similar potency and kinetics?

3. It remains unknown why DEX preferentially affects methylation loci in neutrophils, especially when the average methylation beta values of DEX loci in neutrophils were markedly lower compared with lymphoid cells? Compared with other immune cell subsets, do neutrophils express GR at higher levels? Do they harbor more glucocorticoid responsive elements (GREs), or do they harbor GREs with higher sensitivity?

4. Mechanistic explorations are completely missing for high NDMI-related immunosuppression (reduced CD4+T, increased monocytic MDSC) and poor survival. Gene expression signature in neutrophils and cell-cell interactions should be carefully dissected to rationalize the above correlation.

5. Do different individuals show differences in the "on" and "off" patterns of DNA methylation upon glucocorticoid delivery and withdrawal? It is important to rule out individual variations with sufficient clinical samples. Current data of this part is weak.

6. Tumor progression has been associated with expansion of myeloid cell populations and altered levels of stress hormone glucocorticoid. Although DEX-exposure can elevate NDMI in both glioma cancer patients and non-glioma controls, are there any differences in changes of DNA methylation and immune status (e.g. cell frequency, gene signatures) between these two groups?

Last updated: 2/18/2022

RESPONSE TO REVIEWERS (all reviewer comments are presented in *italics*)

Reviewer #1 (Remarks to the Author): GBM clinical – translational expert

This is an interesting and potentially useful study that describes a blood-based biomarker that can help identify the effect of glucocorticoids (i.e., dexamethasone) on the peripheral immune system in patients with gliomas, who routinely get this immunosuppressive drug. The authors have characterized the DNA methylation profile associated with glucocorticoid administration on a discovery sample set and validated this on a separate set of samples. The investigators used computational deconvolution algorithms that lead them to attribute the steroid-related DNA methylation profiles encountered to neutrophils. They found that this DNA methylation profile correlates with the use of corticosteroids, with decrease in CD4 T cells and the accumulation of monocytic myeloid-derived suppressor cells. This biomarker was associated with shorter overall survival in glioma patients, outperforming the dose of steroids at the time of blood-draw as a prognostic factor. Moreover, in a non-glioma population, the methylation-based biomarker was increased in the context of history of prednisone use. Interesting analysis show non-linear steroid dose to methylation profile dynamics, as well as temporal/longitudinal analysis where the DNA methylation signature of steroids remain for approximately 1 week after stopping this medication. Overall, this biomarker is an elegant approach to characterize the potential iatrogenic immunosuppression that steroids might elicit on glioma patients, an important confounder when evaluating outcomes of glioma related trials, of immunotherapy studies. This is an interesting and valuable study, yet important issues that are worth addressing prior to publication include:

Rev 1 Comment 1: *The deconvolution approach of DNA methylation profile used to attribute the signature encountered to neutrophils is interesting, and the simulation for proportion of neutrophils on the sample adds to the rigor of the approach, but ultimately this was not validated. Proper validation by an independent, complementary experimental approach would be important.*

RESPONSE Rev 1 Comment 1: This reviewer raises an excellent point. To directly respond to this comment, we recruited 5 additional glioma subjects who were taking DEX at the time of blood draw and isolated neutrophils from their whole blood using fluorescence activated cell sorting (FACS). For each of the 5 subjects, DNA methylation was profiled in both isolated neutrophils and in their whole-blood same using the Illumina HumanMethylationEPIC BeadArray, the same array-technology for assessing DNA methylation for all of the other data sets considered in our study. Using this data, we then assessed whether the DNA methylation changes predicted from our bioinformatic approach occur in neutrophils isolated from these five independent glioma subjects. These new validation results are presented in the Results section (lines 478-491) and in the modified Figure 6e and Suppl Table 10. To summarize, our results showed that the NDMI scores of isolated in vivo DEX exposed neutrophils are dramatically increased when compared to the NDMI calculated in isolated neutrophils from non-exposed individuals (n = 6). In each of the 5 independent glioma subjects, the isolated neutrophil NDMI scores are statistically significantly elevated as compared to their corresponding whole blood NDMI scores (P=0.01) (line 485), also consistent with our model. These new results provide independent validation that corroborates our interpretation that neutrophils are highly sensitive to DEX associated DNA methylation and drive the NDMI biomarker of DEX response when examining whole blood. We expanded the Results section (lines 478-491) to discuss our new experiment and other data supporting the validity of our approach. We thank the reviewer and believe the additional work adds rigor to the already robust results.

Rev 1 Comment 2: *To best understand the potential use of this biomarker, comparison of its performance to that of cumulative steroid use, or average use of steroids over weeks to a month, with regards to its prognostic properties in glioma patients is important. In other words, does it make sense to use this sophisticated analysis vs the medication log?*

RESPONSE Rev 1 Comment 2: Our survival studies are based on historical subjects from the UCSF Adult Glioma Study (AGS) for whom medication logs are not available. Data for AGS subjects were derived from our in-person questionnaires asked by epidemiology field interviewers. At the time of the blood draw, patients were only asked if they were currently taking DEX or had taken it in the preceding month. Because patients participated 8 to 30 years ago, many are deceased, and many were treated at other Bay Area institutions.

Thus, obtaining detailed DEX use around the time of surgery and blood draw is not feasible. Our current prospective studies collect relevant DEX dosage but follow-up for these studies is not sufficient to make the survival comparison suggested by Reviewer 1. However, strong support for the methylation approach (NDMI) is provided in a comparison of c-indices from survival models based on the AGS cases where we compared a model with DEX plus prognostic clinical variables (including: age at diagnosis, race, BMI, extent of resection, tumor grade, oligodendroglioma vs. astrocytoma, tumor location and CD4+ T cell proportion) to a model with NDMI plus the same prognostic clinical variables (i.e., the model presented in Table 2 in the revised manuscript). To the left you see the results of splitting the AGS data into a training and test set 20 times. For each time we built the two models on the training sets and then

calculated the c-index for the two models on the respective test sets. It is apparent that the c-index for the NDMI + clinical variable model (as shown in Table 2 in the revised manuscript) is continually higher than that of the DEX + clinical variable model. Please note, we have not included this comparison/figure in the revised manuscript but are happy to do so if the reviewer believes we should.

In addition, in studies of the CD4 and mMDSC counts, detailed data on DEX exposure was collected and analyzed (i.e., cumulative mg, daily mg, duration of exposure), and no DEX dose variable predicted the observed immune parameters as well as the NDMI methylation marker (see Table 1B). We attribute this to the wide and unpredictable individual variability in DEX response, which is captured, at least in part, by the NDMI biomarker. Wide inter-individual variability in corticosteroid response has been noted for many clinical endpoints, and it is also recognized that this cannot be accounted for by drug dosage. This fact is the principal motivation for developing pharmacodynamic markers. DNA level methylation within the highly sensitive neutrophil compartment, as indicated in this work, is one such marker that we believe deserves intensive study. Also, medication logs provide prescribed drug usage, while actual dose and adherence is typically unknown; therefore, an objective biomarker that records actual biologic response could capture information of more significant clinical consequence. Finally, with the use of the online app we will make publicly available (see next comment) and the widespread adoption of DNA methylation arrays, the NDMI is not overly cumbersome or sophisticated and can be easily implemented. The assay requires as little as 0.5 mL of whole blood and can be simply added to routine blood draws.

Rev 1 Comment 3: *The use of this biomarker as a prognostic factor and as a measure of a confounder variable on immunotherapy trials and glioma outcome studies would require a user-friendly interphase. The authors should make an effort to make this analysis algorithm, and training dataset available for its independent validation and implementation and spell out how is this signature best “quantified”.*

RESPONSE Rev 1 comment 3: We have created an online interface (NDMI Shiny App) allowing easy access to the NDMI for researchers (See new section one page 2 of the manuscript). To develop a user-friendly tool for calculating NDMI, R Shiny was used to create an online application URL: <https://btcd.db.shinyapps.io/ndmi/> where users can upload data and calculation of the NDMI score is performed automatically. The portal to the tool is shown in the screenshot to the left. The input for the tool is the DNA-methylation beta-values, subsetted to the 28 NDMI CpGs (listed in Suppl Table 5). The data can be uploaded as an .RData or .csv file where the rows represent the 28 CpGs and each column represents a different sample. The output from the R Shiny NDMI application is a downloadable file containing the NDMI score for each sample. The app instructions will provide guidelines on processing methylation data so that output is most

comparable to the published work. Winston Chang, Joe Cheng, JJ Allaire, Carson Sievert, Barret Schloerke, Yihui Xie, Jeff Allen, Jonathan McPherson, Alan Dipert and Barbara Borges (2021). shiny: Web Application Framework for R. R package version 1.7.1. <https://CRAN.R-project.org/package=shiny>

Rev 1 Comment 4: *The discussion is lengthy, could be streamlined.*

RESPONSE Rev 1 comment 4. Because reviewers raised questions about clinical applications, we have emphasized these at the outset and streamline several portions of the discussion. In response to reviewers we also have elaborated on limitations of the study.

Reviewer #2 (Remarks to the Author): DNA methylation – plasma expert

Major comments

- *A major rewrite is necessary. Results need clear explanation. Figures are not adequately explained and therefore very difficult to properly interpret. Figure legends are inadequate. Conversely, the methods are very long and overly detailed. The paper also lacks a conclusion.*

RESPONSE Rev 2: We have revised legends, shortened methods, and provided conclusions.

- *It is hard to understand the potential application of the NDMI score in a clinical setting. The manuscript needs a better explanation of the main objective of this score. Several times throughout the manuscript, it is proposed that the NDMI score could be used to understand which patients had been exposed to DEX. The rationale behind having an epigenetic signature for DEX exposure is not clear when surely this knowledge is known already. At other instances, the authors imply that the score could be used to understand the effect of DEX in the different populations of immune cells, but the importance of this or potential application is not clear.*

RESPONSE Rev 2: Please see revised Discussion where we now emphasize potential applications of the work.

Rev 2 Comment 1 • *It is unclear whether the EWAS study was adjusted for multiple testing (manuscript only mentions p value, not adjusted p value). If this was not adjusted, it would be advisable to repeat analysis, since this could alter the number of significant differentially methylated probes between DEX exposed and non-exposed patients.*

RESPONSE to Rev 2 Comment 1: We did not adjust our analysis for multiple testing and are grateful for the opportunity to explain our rationale. If our goal was to identify specific CpG loci that are associated with DEX exposure and to have confidence that the discovered loci are indeed differentially methylated between exposed and unexposed individuals, then we completely agree with the reviewer that we would need to do

some type of multiple testing adjustment; otherwise, we are prone to have false positives. However, an EWAS in the traditional sense was not the intent of the first step in our analysis, nor was finding CpG loci that withstand multiple comparison and genome wide significance. The purpose of the EWAS we performed and subsequent CellDMC analysis, was merely to identify “potential” markers of DEX exposure. These “potential markers” were selected deliberately with liberal thresholds on the nominal p-value in order to have sufficient substrate (e.g., putative DEX associated CpGs) in the fitting of the elastic net model, the linear predictor which represents the NDMI score. The statistical significance of our approach is therefore judged by whether the NDMI accurately discriminates DEX users from non-users, not the statistical significance of the individual CpGs that comprise the score. We have checked our revised manuscript to ensure that we do not report individual CpGs as being statistically significant. The focus of our paper is on the NDMI score and not on the individual loci that comprise it. We have added the phrase “nominal significant” to methods and text to emphasize this distinction.

Rev 2 Comment 2 • *GC is a poor choice of abbreviation; it is hard to interpret whether CpG sequences or synthetic glucocorticoids are being discussed. Glucocorticoid is one word and therefore doesn't need an abbreviation!*

RESPONSE to Rev 2. Comment 2: We agree and have replaced the GC abbreviation with glucocorticoid to avoid this confusion.

Rev 2 Comment 3 • *Details on size of training set are missing, line 147-148. Inconsistent details given on line 163: 3,164 total subjects in the AGS, and lines 166-167.*

RESPONSE to Rev 2 Comment 3. We have clarified these details in the revision, lines 186-187 and 200-202.

Rev 2 Comment 4 • *Figure 1A: immune cell deconvolution, survival evaluation set are not explained elsewhere in the manuscript. Overall, very challenging to interpret this figure.*

RESPONSE to Rev 2 Comment 4. We have revised Fig 1 legend to help readers interpret the Figure.

Rev 2 Comment 5 • *Figure 1B is not cited in text.*

RESPONSE to Rev 2 Comment 5. We now cite Fig 1B in the text.

Rev 2 Comment 6 • *Lines 324-325 - B-cells not clearly indicated on either Fig 2 B or C.*

RESPONSE to Rev 2 Comment 6. We added a note to the legend to draw attention to the B cells

Rev 2 Comment 7 • *Figure 2c, color coding of myeloid and lymphoid is not clear to graph,*

RESPONSE to Rev 2 Comment 7. We revised the figures to make the labels larger and more readable.

Rev 2 Comment 8 • *Figure 3 - incorrect labelling in figure legend (first table not described in legend)*

RESPONSE to Rev 2 Comment 8. Thank you, we greatly appreciate this comment. The legend has been revised.

Rev 2 Comment 9 • *Figure 3A: For GR binding sites, is the receptor binding to these sites or is DEX?*

RESPONSE to Rev 2 Comment 9. In these analyses, which are publicly available through ENCODE for readers interested in methodological details, DEX is used to treat cells *in vitro* and following a brief (several hours) incubation, fragmented chromatin is isolated using specific antibodies to the glucocorticoid receptor. The pull-down DNA fragments are then sequenced and mapped to the hg38 version of the human genome. Control (no drug, antibody only) results are compared with the DEX treated pull-downs to designate a chromatin peak as being a “DEX-GR chromatin binding site”. So, the DEX activated glucocorticoid receptor, which contains the bound drug (DEX), is binding to these sites in chromatin.

Rev 2 Comment 10 • *Figure 3C ME1-11 are shown on the x-axis, but in text EM6 is described (line 335), possible typo?*

RESPONSE to Rev 2 Comment 10. Thank you. No there isn't a typo. EM6 is the only module that is enriched among the DEX responsive genes. We clarify this in the revised text.

Rev 2 Comment 11 • *Figure 3E not described in figure legend.*

RESPONSE to Rev 2 Comment 11. Thank you. The legend has been corrected.

Rev 2 Comment 12 • *Prednisone not mentioned in methods, only abstract and results (line 371), data not shown in graphs. How and where were these data collected?*

RESPONSE to Rev 2 Comment 12: For the AGS study we cite the methodology reference # 38 in our methods. Briefly, an in-person interview with each enrolled control or patient was conducted and an extensive list of current and past medications was recorded. The epidemiology study also collected medical, occupational, and dietary information. We have clarified this in the methods. The data in Suppl Fig 2 are presented in Suppl Table 8. A footnote to the Suppl Figure 2 legend refers readers to Suppl Table 8 for data.

Rev 2 Comment 13• *Figure 4, color difference between current and past DEX is difficult to see. In addition, “past DEX” is not clearly defined. What has been the time difference between DEX dose and recording data? Are subjects here plotted twice for previous dose with DEX and follow up Dose of DEX “current DEX”?*

RESPONSE to Rev 2 Comment 13. No subject was plotted twice. We have improved the resolution of Figure 4 to make past DEX folks more visible and we added the days since last DEX to the figure legend. We collected DEX status using two definitions of DEX exposure: current and within the last 30 days. As shown below, the 7 past exposed cases in Figure 4 were last exposed about 3 weeks prior to blood sample, 3 were exposed more recently. Obviously, larger samples with precisely drawn blood samples covering many time points would be optimal to define trends in recovery from DEX but this is very problematic in the context of routine management of brain tumor patients early in their care and prior to surgery. Even though limited, we thought it helpful to show this data, as they provide hints of the kinetic nature of the NDMI. A full evaluation will

require much more data but are not required for the current objectives. The NDMI scores of subjects with past (within 1 month) DEX begin to merge with non-exposed subjects, although not uniformly, suggesting individual variation in recovery from DEX action. We now note the need for further timepoints following DEX in our limitations section of the Discussion. Also, we emphasize many of these training set subjects sampled before surgery were newly diagnosed cases.

Cumulative Dex Dose	7.5	8.0	78.0	23.5	26.0	12.0	22.0
Days from last Dex dose to blood draw	10	22	17	8	18	4	17
Total days took Dex	6	1	12	8	9	1	6
Average Daily Dex	1.25	8	6.5	2.94	2.89	12	3.67
NDMI Score	-4.11	-3.98	-0.95	-3.18	-2.48	0.58	-2.72

This is a highly unique sampling timeframe that avoids any confounding influences of radiation or chemotherapy.

Rev 2 Comment 14• *Figure 4: is this plotting only the training set or training and test? This is unclear, see lines 373-376. This should also be specified in the figure legend.*

RESPONSE to Rev 2 comment 14: Panels A and B (dose response) are only from the training set. In C we also plotted training set data, but in D we used all available data in the test set and additional data, to achieve the largest replication sample size available. The results of C and D are very comparable, which supports the validity of our conclusions. We have added these details in the figure legend. Also see response below. Thank you, this was helpful.

Rev 2 Comment 15• *Lines 396-397. Is this to say that DEX exposure results in a higher methylation level in neutrophils or does it mean that methylation simply varies more?*

RESPONSE to Rev 2 Comment 15. We thank the reviewer for this question and appreciate the opportunity to further elaborate on the interpretation and significance of the results being shown in Figure 4, panels C and D. Figure 4 C, D depict the NDMI (y-axis) as a function of both the predicted relative fraction of neutrophils in whole blood (x-axis) and DEX status (different colored points in the plot). These plots also contain the fitted regression line for DEX+ and DEX- subjects. The fitted regression lines describe the linear relationship between the predicted relative fraction of neutrophils in whole blood and NDMI and were estimated separately for DEX+ and DEX- subjects. Finally, the p-value being reported in Figure 4 C, D is based on a test of equality of slopes between the regression lines fit to the DEX+ and DEX- subjects. Testing the equality of slopes is to assess whether the linear relationship between the predicted relative fraction of neutrophils in whole blood and

NDMI are the same between DEX+ and DEX- subjects. As the p-values for the tests of equality of slopes are statistically significant ($p < 0.05$) in both the training data set (Figure 4C) and in an independent data set of 99 DEX exposed glioma subjects and 453 non-exposed control subjects (Figure 4D), this strongly suggests that the linear relationship between predicted relative fraction of neutrophils in whole blood and the NDMI is different between DEX+ and DEX- subjects. More specifically, the slope is greater among those exposed to DEX. Thus, while the NDMI appears to increase as a function of the relative fraction of neutrophils in both DEX+ and DEX- subjects (Figure 4 C, D), the fact that it increases at a higher rate among DEX+ subjects support our interpretation of the results generated via CellDMC and suggests enhanced sensitivity to DEX exposure within the neutrophil compartment. If the slope gradients of DEX exposed and non-exposed subjects were not different, then that would refute our interpretation, however, that was not observed. Instead, the plots reveal the expected outcome of the linear regression models evaluated in the CellDMC analyses. We have edited the legend for Figure 4 to include a more thorough description of subpanels C and D.

Rev 2 Comment 16 • Fig 5A: *non-cell specific score and cell proportion scores need proper explanation*
RESPONSE to Rev 2 Comment 16. We explain the creation and definition of these scores in methods section (lines 241-260).

Rev 2 Comment 17 • *The text does not specifically reference Fig 6A, B or C. P values should be added to the graphs*
RESPONSE to Rev 2 Comment 17. We have modified the text to refer to Fig 6 and have added p-values to all the graphs

Rev 2 Comment 18 • *Figure 7 GBM, IDH abbreviations in legend not defined elsewhere in text.*
RESPONSE to Rev 2 Comment 18. Abbreviations are defined in the Methods section, lines 209-210.

Rev 2 Comment 19 • *Lines 315-316: predominant effect of DEX is hypomethylation. Needs more explanation.*
RESPONSE to Rev 2 comment 19. The definition of hypomethylation is described and modified in lines 637-638 of the Figure Legend 2A.

Rev 2 Comment 20 • *Line 456: Are the pre-surgery subjects glioma patients? Needs clarification.*
RESPONSE to Rev 2 comment 20. Yes, all the presurgery subjects are glioma subjects; this is now stated at several points, including in the Figure 1 legend, and Methods section.

Rev 2 Comment 21 • *Lines 500-502: These are not shown, results do not go through any data on chromatin accessibility and DNA methylation.*
RESPONSE to Rev 2 comment 21. We now describe chromatin results in the Results section lines 390-392.

Rev 2 Comment 22. *Line 515: Mention that they looked at histone methylation, but no results have been shown where they conducted this analysis.*
RESPONSE to Rev 2 comment 22. We describe the eFORGE analysis in Methods and Results section (lines 276-277 and 390-392).

Rev 2 Comment • *Methods (lines 216-217): There is mention of an external dataset with 311 DEX- control samples. It is unclear whether these are also glioma samples.*
RESPONSE to Rev 2 Comment 23. We greatly appreciate this comment. The 311 controls were a typo and has been corrected. The DEX- control samples are described in Figure 1 and the sample size is n=453 subjects. The term “external” was poorly chosen, the term “independent”, which is now used, better describes this group.

Rev 2 Comment 24 • *Results + Figure 7: Authors should take advantage of the number of patients in the test dataset to do survival analysis. Instead, they use hypothetical patients (only a few) to show correlation between NDMI scores and survival prognosis. This might lead to a biased interpretation of the data and its potential use in the clinical setting.*
RESPONSE to Reviewer 2 Comment 24: We apologize for the confusion. In fact, we did use the entire test dataset for the survival analysis. We built a multivariate Cox proportional hazards model with the test dataset (described on lines 323-331). Plotting survival curves from a Cox model differs from the Kaplan-Meier method.

The latter is inherently univariate, and, thus, does not account for any other variables. When plotting the survival curves for a specific variable (here, NDMI) from a multivariate Cox model, the other variables (age, surgery, race, BMI, and CD4T cells) in the model are fixed to their average or mode values (e.g., age is set as the mean age for the subset of patients (IDH-WT GBM or IDH-WT Astro)) and the interpretation is similar to that in a multivariate linear model. Thus, the hypothetical patients are just the test dataset average (mode) values for age, surgery, race, BMI, and CD4T cells with the observed test dataset NDMI and survival times. Considering the reviewers' comments, and for a more easily interpretable visualization, we have now replaced the Cox model survival curves with the Kaplan-Meier curves for visualizing the quartiles of the NDMI in the IDH-WT GBM and IDH-WT Astrocytoma patients (see new Figure 7).

Reviewer #3 (Remarks to the Author): DNA methylation – liquid biopsy expert

The authors present a novel blood-based biomarker, coined NDMI, that provides a measure of epigenetic response of subjects to dexamethasone. Dexamethasone is a commonly used steroid with a wide range of side effects and there exist few ways of predicting an individual's response. A few questions that may strengthen the manuscript:

Rev 3 Comment 1) *The use of dexamethasone in the pre-, peri-, and post-operative management of glioma is an important, and debated, clinical question. While NDMI may be a prognostic factor, it remains unclear to me how this would change the care of patients with glioma. If a neuro-oncologist is provided with a patient's NDMI score, what would they do differently for a given patient? If a patient is symptomatic with significant vasogenic edema and an NDMI score that suggests poor dexamethasone response, there are few alternatives.*

RESPONSE Rev 3. Comment 1: Please see revised discussion. Currently DEX is often given before onset of symptoms, or sometimes before and especially following surgery in a prophylactic manner. An alternative to prophylactic treatment could be to wait until symptoms present indicating need for instituting the drug. Also, lower doses could be used and titrated upwards more cautiously. Furthermore, there is no consensus on the dosages used in neuro-oncology, although a typical course in our institution begins with a high daily dose (16 mg) that is tapered over the course of about 2 weeks. There is a lack of data in this field. Finally, there are alternative therapies such as bevacizumab that can be instituted early and act in a "steroid sparing" fashion. The latter treatment itself poses risks for adverse effects. Thus, the reviewer is correct that in symptomatic patients there are few alternatives, but dose reductions and substitutions are available. A current clinical trial is underway in Europe (NCT04266977) to assess the safety of a restricted use of DEX in GBM patients to provide some hard numbers regarding the risks of much more conservative use of the drug. Even if the use of steroids is unavoidable in some patients, their individual response to the drug (in terms of immune cell epigenetics) could aid in prognostic risk stratification and selection for clinical trials, particularly those targeting immune modulation. Finally, while not a focus of this study, mapping the epigenetic responses of DEX may help elucidate gene pathways involved in the drug's CNS vasogenic response and help efforts to develop more targeted edema reducing alternatives that do not have the immunosuppressive effects that DEX produces.

REVIEWER 3 Comment 2. *What is the performance of the NDMI score if built and validated on GBM samples alone, the patients in whom prolonged dexamethasone treatment is most likely? The other brain tumor patients are less likely to require prolonged dexamethasone use.*

RESPONSE Rev 3 Comment 2. In response to this we have constructed an NDMI predictor just using the GBM samples only. We then evaluated the performance of this predictor in the training and test sets. We then compared the GBM-only predictor to the original NDMI and cell proportion predictors using receiver operator ROC analyses. The results (presented in revised Figure below) indicate that although the GBM-only predictor performs well it does not discriminate as well in the test data as the original marker using all grades of glioma did. Therefore, we feel the original NDMI algorithm is superior and would also note we do have grade II and III subjects receiving DEX in the study. Lower grade tumor patients, particularly at recurrence, receive DEX. The ROC curves clearly indicate that the performance of the GBM only NDMI model (light blue curve) is comparatively lower as compared to the performance of the original NDMI model (dark blue curve). The original NDMI model has an AUC of 91% (CI 87.1%-95%) to predict DEX+ Glioma cases and DEX- Control samples as compared to the GBM only NDMI model which has an AUC of 87.87% (CI 83.4%-92.3%). We appreciate the comment but feel the original model is superior and is the one we have created the online app for users to calculate the NDMI from their own methylation data.

REVIEWER Rev 3 Comment 3: *Key clinical variables are not accounted for in the survival analysis: tumor volume, tumor location, and extent of resection (biopsy vs. any resection is not ideal).*

RESPONSE Rev 3 Comment 3: Thank you for the comment. In the AGS data only 80 pre-surgery tumor volumes are available. Due to the nature of a Cox Proportional Hazards model, any missing variable values (e.g., tumor volume) drops the patient from the analysis. Thus, we have not included tumor volume. Unfortunately, the only extent of resection on this cohort is biopsy vs. any resection. We agree this is not ideal; however, we cannot extract additional information. However, based on your suggestions, we were able to further extract tumor location and have now included it as a variable. The updated model is shown in the revised Table 2. Please note, locations that were not statistically different in survival were combined (i.e., the baseline variable includes tumors in the Frontal Lobe/Occipital Lobe/Cerebrum/Other). The main conclusions regarding the NDMI score were unaffected by the addition of tumor location.

Rev 3 Comment 4) *While preclinical data and retrospective studies, as cited by the authors, the association between dexamethasone and poor survival in GBM has not been tested in an RCT. This should be noted.*

RESPONSE Rev 3 Comment 4: This is a good point. To our knowledge, no RCT has ever tested DEX specifically. We make note of this fact in our revised background section (lines 151-153). We also elaborated on the limitations of our knowledge and the current study and specifically note this point. We cite a current RCT NCT04266977, that aims to test whether restricted DEX use adversely impacts GBM outcomes.

Rev 3 Comment 5) *Figure 7A and 7B, the text and legend describe the survival curves for IDH WT GBM and IDH WT astrocytoma, respectively. However, the number at risk tables for both 7A and 7B are the same. Please clarify.*

RESPONSE Re 3 Comment 5: Please see response above to Reviewer 2 Comment 24. We have replaced our Figure 7 with a Kaplan Meier plot and present a revised Cox model in Table 2. The numbers of subjects are depicted correctly for IDH-WT GBM and IDH-WT astrocytoma.

Rev 3 Comment 6) *Increasing number of studies have shown that many IDH WT astrocytoma have very similar survival to GBM. Why is there not a statistically significant association between survival in IDH WT astrocytoma in Table 2 and Figure 7?*

RESPONSE Rev 3 comment 6. In Table 2, IDH-WT astrocytoma are the baseline comparator for the other groups which is why it has a hazard ratio of 1 and no 95% CI or p-value. The table shows that IDH-WT GBM compared to IDH-WT astrocytoma are significantly different in risk of death (IDH-WT GBM have a higher risk than IDH-WT astrocytoma (HR=1.79)). Oligodendrogliomas compared to IDH-WT astrocytoma are also significantly different in risk of death (oligodendrogliomas have a lower risk than IDH-WT astrocytoma

(HR=0.71)). So, our data is compatible with the reviewer's expectations based on current literature. The model presentation is a standard statistical format but can be confusing at first glance. We have added a footnote to highlight that IDH-WT astrocytoma are the baseline for the model. Thank you for the comment.

Rev 3 Comment 7) *The authors comment that there is substantial variability in NDMI scores in glioma patients and this variability is not explained by dexamethasone dose or exposure. This variability is quite notable in Figure 4. They suggest that pre-drug methylation status is an important variable. How do the authors propose accounting for this in clinical practice?*

RESPONSE Rev 3 Comment 7. This is an excellent point and indeed relevant if the change in NDMI score is to be used as the predictor variable. Obviously, to be implemented in this way one would require a baseline (pre-DEX) measurement. Because patients are often treated before referral to our institution such a baseline study would be problematic. To make full use of biomarkers some change in awareness concerning glucocorticoid use may be needed both to limit the dose when feasible and to conduct the baseline studies. However, the change in NDMI following first course of DEX may not be a necessary variable. NDMI taken before, after and during DEX as in the current study was predictive of blood CD4 and mMDSC counts. In the survival studies the NDMI was a prognostic factor even though it was measured at various times during and post DEX. Longitudinal studies comparing various timepoints are necessary to optimize the use of the NDMI tool. The current scope of study was not powered to take on a full evaluation of the NDMI marker and we note this in the limitations section (lines 594-610) and call for more extensive longitudinal studies including inclusion in future clinical trials.

Rev 3 Comment 8) *What conditions did the control individuals exposed to prednisone have? Would be very interesting if the authors could test other inflammatory conditions in the brain treated with steroids as controls.*

Response Rev 3 Comment 8. We now include a listing of the conditions among controls (Lines 407-409). Medical conditions recorded among the 18 non-glioma controls receiving oral prednisone included a diverse group: 6 rheumatoid arthritis, 6 solid organ transplants (2 kidney; 4 not specified), 1 lupus erythematosus, 1 scleroderma, 1 multiple sclerosis, 1 uveitis, 1 hepatitis and 1 connective tissue disorder not specified. We thank the reviewer for this comment as these details point out the many potential applications of our biomarker including in neuroinflammatory disease.

Reviewer #4 (Remarks to the Author): glucocorticoid in cancer expert

Dr. Kelsey and colleagues investigated the relevance of glucocorticoid therapies and DNA methylation remodeling in peripheral immune compartment. With tools and strategies including epigenome-wide association, CellDMC, immune cell deconvolution, and elastic net regression, they developed a biomarker for dexamethasone (DEX) response, called neutrophil DEX methylation index (NDMI). By data mining of the UCSF Immune Profiles Study and the San Francisco Adult Glioma Study, used as the training and test cohort respectively, they show that NDMI exhibited prognostic values for the survival of glioblastoma and astrocytoma. Also, high NDMI was found to correlate with depressed CD4 T cells and elevated monocytic MDSC in DEX-treated individuals, indicating an immunocompromised status. Thus, this study proposed NDMI as a highly sensitive and specific epigenetic biomarker for current DEX-exposure with prognostic values of glioma survival. In general, several major concerns should be carefully addressed to consolidate the conclusions.

Rev 4 Comment 1. *Can major conclusions be extrapolated to other cancer types? Does NDMI only exhibit prognostic values in certain subtypes of glioma?*

RESPONSE Rev 4 Comment 1. The NDMI was a significant predictor of survival in our AGS study among the major subtypes of adult glioma, i.e., IDH-WT GBM and IDH-WT astrocytoma, but not the more uncommon oligodendroglioma (Oligo). The use of DEX is far less frequent in Oligo patients and their survival times are much longer than astrocytic glioma, hence there are issues concerning the power of our studies in Oligo patients. Although the frequency of DEX use varies by subtype, the NDMI was sensitive to DEX exposure in all subtypes in our training and test data. Therefore, our main conclusion is that the NDMI is a sensitive marker

of DEX response in all types of glioma but may be predictive of survival in the major subsets that have the greatest DEX exposure at the time of blood sampling. There are no conclusions we can make concerning other cancer types at this time and this is beyond the scope of this report. We are examining the NDMI in various cancer cohort studies and will report on those findings in the future.

Rev 4 Comment 2. *Tumor progression can augment the production of endogenous glucocorticoid and activate GR signaling. Do high levels of endogenous glucocorticoid correlate with increased NDMI in glioma cancer patients? Do different types of synthetic glucocorticoids alter NDMI with similar potency and kinetics?*

RESPONSE Rev 4 Comment 2. We appreciate the question and agree these are interesting subjects for future research. Unfortunately, given the ethical constraints against experiments with high doses of these drugs it is unlikely that truly comparable data can be collected in humans. We are working towards analogous methylation indices that can be applied to animals. Such comparative pharmacodynamic studies will have to wait until these tools are available. Our limited data in oral prednisone use among non-glioma subjects indicates that commonly prescribed treatments of that drug in an outpatient setting are sufficient to register a significant NDMI response. We also point out that inhaled corticosteroid use was not associated with elevated NDMI, which is consistent with the literature showing limited systemic exposure via the inhalation route. This crude understanding of relative potency will have to suffice until future research is completed but does not detract from our primary conclusions in glioma.

Rev 4 Comment 3. *It remains unknown why DEX preferentially affects methylation loci in neutrophils, especially when the average methylation beta values of DEX loci in neutrophils were markedly lower compared with lymphoid cells? Compared with other immune cell subsets, do neutrophils express GR at higher levels? Do they harbor more glucocorticoid responsive elements (GREs), or do they harbor GREs with higher sensitivity?*

RESPONSE Rev 4 Comment 3: The reviewer raises questions concerning the most fascinating aspect of the current findings. The neutrophil has been relatively understudied as a cell of interest in immunobiology and commonly regarded as an end stage terminally differentiated “delivery system” for bactericidal products. Instead, very recent work shows this lineage to be highly plastic with potential for immunoregulatory properties (Sinha S 2022 Nat Med). We note that there are no glucocorticoid receptor binding chromatin maps for human neutrophils as there are for monocytes and T or B lymphocytes. To our knowledge there is only one single cell RNA seq study in the literature using human neutrophils that examined glucocorticoid effects. We reference this in our revised discussion (lines 608-610, ref # 37). We discuss the severe gaps in our knowledge in the discussion. We highlight one important work (ref #36) that reveals the high sensitivity of neutrophils to glucocorticoid induced transcriptional response. We can only highlight the lack of study of this cell type and emphasize the need for more research. As for GRE and receptors, there are early studies showing somewhat higher numbers of GR alpha and beta in neutrophils, but we think these are unlikely to explain the effects that drive transcription and DNA methylation.

RESPONSE Rev 4 Comment 4. *Mechanistic explorations are completely missing for high NDMI-related immunosuppression (reduced CD4+T, increased monocytic MDSC) and poor survival. Gene expression signature in neutrophils and cell-cell interactions should be carefully dissected to rationalize the above correlation.*

RESPONSE Rev 4 Comment 4: While we agree mechanistic studies are needed, we do not agree that they are crucial for our purpose of creating a biomarker of glucocorticoid response that is bridged to clinically relevant endpoints, the latter being the goals of the study. The empirical associations we report show that the NDMI is a highly accurate reflection of DEX response (outperforming immune cell distributions) and that it predicts two critical immunosuppression parameters (CD4 and mMDSC counts) as well as patient survival in the multivariate setting. We would argue that the current correlative data is a prerequisite that lays a foundation for motivating such mechanistic studies, which will require bench level science and potentially clinical trials to be successful. We cite the need for more mechanistic work in our Discussion limitations section (lines 594-611).

Rev 4 comment 5. *Do different individuals show differences in the "on" and "off" patterns of DNA methylation*

upon glucocorticoid delivery and withdrawal? It is important to rule out individual variations with sufficient clinical samples. Current data of this part is weak.

RESPONSE Rev 4 Comment 5. No one CpG site functions in an “on” or “off” pattern to indicate glucocorticoid action. Instead, the NDMI is the linear component of an elastic net predictor that sums across 28 different CpG sites, each weighted by a coefficient that maximizes discrimination of DEX exposed versus non-exposed subjects. Considering this, the reviewer’s question poses several challenges. As for the kinetics of recovery following removal of drug, this is a complex experiment to operationalize. This is because steroid treatments in neuro-oncology are not standardized for dose and involve a dynamic dosing pattern (taper). i.e., they begin with a high daily dose and gradually taper to a low dose. Access to patients during these treatments is limited to scheduled blood draws that do not coincide with DEX treatment schedules. In addition, patients receive radiation and chemotherapy following surgery, then adjuvant chemotherapy, and these treatments span about 6 months, also not coordinated with DEX exposure. Access to patients is limited and these concurrent treatments make longitudinal interpretation quite challenging. However, we are collecting such data, but it will take several years to make meaningful conclusions. We offer some preliminary data but given the constraints of real-life patient management such data are very difficult to collect; doing any kind of experiment involving alteration of clinically indicated DEX dosing is obviously inappropriate in the hospital setting.

As for the reviewer’s query about individual differences in the patterns of DNA methylation, we agree that this is an interesting question, but we do not have the ability to explore this at any level of granularity. The performance of the NDMI in ROC analyses and its associations with important immune variables (CD4 and mMDSC counts) as well as patient survival are the benchmarks, we use to judge the consistency of this biomarker across patients. An additional point to keep in mind is that corticosteroids and other nuclear hormone receptor mediated effects on chromatin do not follow an “on” and “off” pattern. Binary methylation changes are limited to developmental and differentiation processes during the creation of distinct cell lineage subtypes. Instead, cortisol and synthetic glucocorticoids act on “preprogrammed” active chromatin sites in the genome; they do not act as pioneering transcription factors. We allude to this distinction in the introduction. Hence, DEX has the effect of reinforcing and amplifying already active chromatin. We attempt to illustrate this in Figure 2b, which shows that CpGs modified by DEX are sites that are already relatively demethylated (open chromatin), even in non-exposed cells. The graded (non-binary) nature of glucocorticoids on DNA methylation makes the reviewer’s question more complex to test, however, we agree this a logical concern. We acknowledge these limitations in our discussion.

REVIEWER 4 Comment 6. *Tumor progression has been associated with expansion of myeloid cell populations and altered levels of stress hormone glucocorticoid. Although DEX-exposure can elevate NDMI in both glioma cancer patients and non-glioma controls, are there any differences in changes of DNA methylation and immune status (e.g., cell frequency, gene signatures) between these two groups?*

RESPONSE Rev 4 Comment 6. In response to the reviewers question we have compared the immune cell proportions of control subjects taking prednisone to those controls not taking prednisone and to glioma patients. The results indicate that prednisone elevates the NDMI in controls and alters the distributions of immune cells in ways very similar to DEX exposure in glioma subjects. The results are shown below in Suppl

Supplementary Table 9. Immune Profile Comparisons by Glucocorticoid Use at Blood Draw, Adult Glioma Study Controls

Glucocorticoid use at blood draw	Neutrophil proportions	NLR	CD4 proportions	CD8 proportions	CD4/CD8 Ratio	NK cell proportions	B cell proportions	Monocyte proportions	Total Lymphocytes
No GC exposure (n=412), Median (IQR)	57.8 (51.1, 65.3)	1.6 (1.2, 2.2)	15.7 (12.0, 19.9)	8.8 (6.2, 12.5)	1.8 (1.2, 2.6)	4.7 (3.5, 6.2)	5.3 (4.0, 7.0)	7.4 (6.0, 9.3)	37.2 (29.5, 43.6)
Oral prednisone use (n=18), Median (IQR)	63.9 (59.1, 81.8)	2.1 (1.7, 5.3)	9.7 (6.4, 14.4)	8.2 (4.2, 11.3)	1.2 (1.0, 1.9)	3.9 (2.0, 5.2)	4.1 (3.0, 4.6)	7.1 (6.1, 9.8)	30.3 (15.6, 35.1)
Inhaled GC use (n=9), Median (IQR)	62.9 (60.8, 67.4)	2.1 (1.7, 2.2)	15.5 (13.6, 17.1)	4.9 (4.3, 12.5)	3.3 (1.2, 3.6)	3.2 (3.0, 4.5)	3.8 (3.0, 6.0)	7 (6.0, 8.2)	30.3 (29.4, 36.4)
P-value (Linear ANOVA comparing differences between all 3 groups)	< 0.001	< 0.001	< 0.001	0.56	0.28	0.08	0.04	0.85	< 0.001
Values in Prednisone exposed versus controls with no GC exposure	Increased	Increased	Decreased	Not statistically different	Not statistically different	Decreased	Decreased	Not statistically different	Decreased

Table 9. In other words, in control prednisone users, the proportions of neutrophils and the neutrophil lymphocyte ratio are increased and the proportions of CD4, CD8 and B-cells are decreased compared to non-users. We are limited in drawing specific conclusions but the pattern of changes in controls using prednisone and glioma

subjects taking DEX are highly similar. We do not have access to gene expression data, although that would be interesting. Future studies are planned to assess gene expression and DNA methylation in the same samples. We hope the current study and the app we have created for easy calculation of the NDMI score will

enable other researchers to explore this biomarker in different glucocorticoid exposures in a variety of clinical contexts.

REVIEWERS' COMMENTS

Reviewer #1 (Remarks to the Author):

While this is a hypothesis generating study, I think overall the manuscript describes an interesting, novel approach to determining the effect of corticosteroids on the circulating immune compartment of brain tumor patients. The authors have been responsive, and reasonably addressed my critiques, and include a user-friendly online interphase that will hopefully allow other groups to independently apply this biomarker to other datasets to eventually validate or disregard the use and value of this tool. Therefore, I would recommend this manuscript for publication.

Reviewer #3 (Remarks to the Author):

I thank the authors for their careful revisions. Overall, the manuscript is improved may benefit from further revision, particularly with contextualizing the clinical applications of current study and controlling for known predictors of glioma survival (eg tumor volume and extent of resection)

1. While the authors' discussion regarding the utility of NDMI in immunotherapy clinical trials is likely appropriate, I remain curious about how NDMI would be incorporated into clinical practice. Considering the case of a symptomatic newly diagnosed GBM patient, when would NDMI change management? The authors mention bevacizumab - though it appears unlikely this would be used in such a patient. While the discussion is improved, it would strengthen the manuscript to further characterize how NDMI may be integrated into neuro-oncologic care
2. I thank the authors for examining NDMI development in GBM samples only. Given NDMI is likely of greatest utility in the GBM population, newly diagnosed and recurrent, this result should be presented as a supplementary finding.
3. The inability to control for tumor volume and extent of resection represents a significant limitation in the survival models. Are there validation data sets available that would allow calculation of NDMI scores and controlling for tumor volume and extent of resection? Validation data sets would add significantly to the manuscript.

Reviewer #4 (Remarks to the Author):

The authors aimed to develop a biomarker (called NDMI score) for glucocorticoid response and glioma survival. In general, the authors haven't made much efforts to address major concerns raised by reviewers. Thus, the quality and reliability of this study are still below the expected average bar of Nature Communications. Mechaistic explorations are still completely missing. Although the authors may have some limitations to clinical samples, some ex vivo studies or in vivo murine studies should be feasible.

We have addressed all of the Reviewer comments in the revised manuscript and respond with additional data and experiments that will appear as supplemental data.

Below reviewers' verbatim comments are italicized.

REVIEWER 1. Comment

While this is a hypothesis generating study, I think overall the manuscript describes an interesting, novel approach to determining the effect of corticosteroids on the circulating immune compartment of brain tumor patients. The authors have been responsive, and reasonably addressed my critiques, and include a user-friendly online interphase that will hopefully allow other groups to independently apply this biomarker to other datasets to eventually validate or disregard the use and value of this tool. Therefore, I would recommend this manuscript for publication.

RESPONSE to REVIEWER 1. Thank you. We agree that additional studies will ultimately determine the value of the NDMI biomarker.

REVIEWER 3. Comment

I thank the authors for their careful revisions. Overall, the manuscript is improved may benefit from further revision, particularly with contextualizing the clinical applications of current study and controlling for known predictors of glioma survival (e.g., tumor volume and extent of resection)

1. While the authors' discussion regarding the utility of NDMI in immunotherapy clinical trials is likely appropriate, I remain curious about how NDMI would be incorporated into clinical practice. Considering the case of a symptomatic newly diagnosed GBM patient, when would NDMI change management? The authors mention bevacizumab - though it appears unlikely this would be used in such a patient. While the discussion is improved, it would strengthen the manuscript to further characterize how NDMI may be integrated into neuro-oncologic care

2. I thank the authors for examining NDMI development in GBM samples only. Given NDMI is likely of greatest utility in the GBM population, newly diagnosed and recurrent, this result should be presented as a supplementary finding.

3. The inability to control for tumor volume and extent of resection represents a significant limitation in the survival models. Are there validation data sets available that would allow calculation of NDMI scores and controlling for tumor volume and extent of resection? Validation data sets would add significantly to the manuscript.

RESPONSE to REVIEWER 3:

In response to 1:

We agree that further study is needed to fully establish the ways NDMI can be incorporated into clinical practice beyond the important potential case of risk-stratifying patients for immunotherapy clinical trials. While the NDMI doesn't change the need for steroids in symptomatic GBM patients, calculation of a patient's sensitivity to steroids could be used to calibrate the dose given to an individual patient with the dual goals of providing adequate clinical efficacy and minimizing toxicity, in particular if there is a need to re-start steroids during therapy. Such a "precision" approach has the potential to significantly reduce steroid-related morbidity in the GBM patient population. We have edited the discussion to try to emphasize these points.

The prognostic impact of all immune cell profiles (including ratios and cell subtypes) are increasingly being investigated, and all are inherently confounded by the impact of glucocorticoids. Further validation of NDMI as a more quantitative marker of response to glucocorticoids may allow for more

accurate interpretation of cell subtypes and ratios on prognosis, response to treatments, and other clinical outcomes.

In response to 2. We have added the analyses using only GBM in the revised SUPPLEMENTAL results as suggested by the reviewer.

In response to 3. In response we have obtained additional data and carried out analyses as requested by Reviewer 3. Preoperative and postoperative tumor volumes were available for a subset of our AGS cohort (n=74). Using blood NDMI and immune cell estimates and clinical variables, we evaluated multivariate Cox proportional hazards survival models including NDMI and extent of resection with and without pre-operative tumor volume. Importantly, in both models, the NDMI score remained statistically significant as a predictor of survival. In both models, volumetric measurements were significant as were age and WHO tumor classification. DEX use at blood draw was not statistically significant in either model (Supplemental Tables 11-12). These data do not support a confounding role of tumor volume or extent of resection on the association of NDMI with glioma survival. Instead, they support our interpretation that the NDMI captures information of importance in glioma survival that is not captured by the history of DEX use at the time of blood draw.

REVIEWER 4 comments

The authors aimed to develop a biomarker (called NDMI score) for glucocorticoid response and glioma survival. In general, the authors haven't made much efforts to address major concerns raised by reviewers. Thus, the quality and reliability of this study are still below the expected average bar of Nature Communications. Mechanistic explorations are still completely missing. Although the authors may have some limitations to clinical samples, some ex vivo studies or in vivo murine studies should be feasible.

RESPONSE to REVIEWER 4: We respectfully disagree that mechanistic explorations are completely missing. We propose that our studies of immunosuppressive MDSCs and depletion of vital T cell populations are a mechanistic insight into how the NDMI score may be associated with patient survival as these immune factors are known *a priori* to be associated with poor survival. In our first revision, we also carried out isolations of neutrophils from glioma patients to bolster our interpretation that a mechanism targeting neutrophils drives the association of DEX and DNA methylation in whole blood. We believe this provides insight into mechanism. As for mechanistic insights into the molecular modification responsible for changes in DNA methylation in human cells by glucocorticoids, we believe this complex topic is beyond the scope of a clinic-based study. The DNA methylation arrays for mice are still inadequate to test these ideas.